# miR-29b contributes to multiple types of muscle atrophy

Jin Li[1,*], Mun Chun Chan[2,3,*], Yan Yu[4,*], Yihua Bei[1], Ping Chen[1], Qiulian Zhou[1], Liming Cheng[4], Lei Chen[4], Olivia Ziegler[2], Glenn C. Rowe[5], Saumya Das[2] & Junjie Xiao[1]

A number of microRNAs (miRNAs, miRs) have been shown to play a role in skeletal muscle atrophy, but their role is not completely understood. Here we show that miR-29b promotes skeletal muscle atrophy in response to different atrophic stimuli in cells and in mouse models. miR-29b promotes atrophy of myotubes differentiated from C2C12 or primary myoblasts, and conversely, its inhibition attenuates atrophy induced by dexamethasone (Dex), TNF-α and $H_2O_2$ treatment. Targeting of IGF-1 and PI3K(p85α) by miR-29b is required for induction of muscle atrophy. _In vivo_, miR-29b overexpression is sufficient to promote muscle atrophy while inhibition of miR-29b attenuates atrophy induced by denervation and immobilization. These data suggest that miR-29b contributes to multiple types of muscle atrophy via targeting of IGF-1 and PI3K(p85α), and that suppression of miR-29b may represent a therapeutic approach for muscle atrophy induced by different stimuli.

[1] Cardiac Regeneration and Ageing Lab, School of Life Science, Shanghai University, Shanghai 200444, China. [2] Cardiovascular Division of the Massachusetts General Hospital and Harvard Medical School, Boston, Massachusetts 02215, USA. [3] Biology Department, Georgetown University, Washington DC 20007, USA. [4] Department of Spine Surgery, Tongji Hospital, Tongji University School of Medicine, Shanghai 200065, China. [5] Department of Medicine, The University of Alabama at Birmingham, Birmingham, Alabama 35294, USA. * These authors contributed equally to this work. Correspondence and requests for materials should be addressed to J.X. (email: junjiexiao@shu.edu.cn).

Muscle atrophy is a debilitating systemic response to denervation, long-term inactivity, excessive fasting, ageing, and a variety of diseases including excessive glucocorticoids (as in Cushing syndrome) and cancers[1,2]. Muscle atrophy can lead to poor functional status, reduced quality of life, and increased morbidity and mortality[3]. The treatment of muscle atrophy remains an unresolved challenge[3]. Therefore, there is an urgent unmet need for the development of novel therapies to combat loss of skeletal muscle mass[1,4,5].

Inhibition of the insulin-like growth factor 1-phosphoinositide 3-kinase-AKT (IGF-1–PI3K–AKT) signalling pathway has been implicated in the induction of muscle atrophy[6–8]. Inhibition of AKT leads to the activation of transcription factor Forkhead Box O3 (FOXO3). FOXO3A induces increased expression of atrophy-linked ubiquitin ligases, including muscle-specific RING-finger 1 (MURF-1) and Atrogin-1 (MAFBX)[1,6].

MicroRNAs (miRNAs, miRs) constitute a class of highly conserved, small endogenous noncoding RNA molecules that negatively regulate gene expression at the posttranscriptional level[9,10]. Individual miRNA can target several mRNAs, while a single mRNA can be regulated by a variety of miRNAs[11–13]. Thus, miRNAs have been reported to play fundamental roles in diverse biological and pathological processes, including muscle development and regeneration[14–16]. Different miRNAs, including miR-1, miR-133 and miR-206 have been shown to participate in myogenesis and muscle regeneration[14–16]. In contrast, miRNAs have also been shown to play a role in different models of muscle atrophy, including miR-1, miR-133, miR-23a, miR-21, miR-27, miR-628, miR-431 and miR-206 (refs 17–24). However, a systematic study to examine the role of miRNAs using different models of muscle atrophy has not been performed.

In this study, using an miRNA array, we identify miR-29b induction and activity as a novel pathway contributing to muscle atrophy. miR-29b expression is significantly increased in multiple in vivo and in vitro models of muscle atrophy. We also confirm IGF-1 and PI3K(p85α) as two target genes of miR-29b. Finally, we demonstrate that miR-29b expression is necessary and sufficient to induce muscle atrophy in vivo. These data suggest that inhibition of miR-29b might represent a novel therapeutic approach for multiple types of muscle atrophy.

## Results

**miR-29b is increased in multiple types of muscle atrophy.** To identify miRNAs that play a role in muscle atrophy, miRNA arrays were performed on the gastrocnemius muscles from rats that have undergone denervation of the right sciatic nerve (Fig. 1a–e). A relative early time point (day 5) was used here to identify miRNAs that might function as potential triggers of muscle atrophy. Expression levels of 15 miRNAs were found to be changed in denervated muscle compared to control (Fig. 1e and Supplementary Table 1). A total of eight miRNAs with the largest inductions were technically validated using quantitative real-time polymerase chain reactions (qRT–PCRs) (Fig. 1e). Among them, miR-212, miR-29b, miR-21 and miR-221 were confirmed to be increased in both rat and mouse denervated gastrocnemius muscles (Fig. 1e).

To explore whether the identified miRNAs could play a role in other models of muscle atrophy, their expression levels were analysed in four additional in vivo models of skeletal muscle atrophy (Fig. 1f–i and Supplementary Fig. 1a–e), namely muscle atrophy induced by dexamethasone (Dex), fasting, cancer cachexia and ageing. Of the four validated miRNAs in denervated muscles, only miR-29b was found to be elevated in each of the in vivo atrophy models (Fig. 1f–i). To investigate the time course of miR-29b expression in denervated muscles, we checked its

expression level at 3, 5, 7 and 14 days after denervation and found that miR-29b was induced at day 5 and maintained at higher levels after (Supplementary Fig. 1f). Besides that, to explore if the upregulation of miR-29b in denervation was specific in gastrocnemius muscles or it was a more generalized process, we checked miR-29b expression level in other muscles after denervation including tibialis anterior (TA), soleus and extensor digitorum longus (EDL), and found that it was consistently elevated in all these denervated muscles (Supplementary Fig. 1g).

To further explore the regulation of miR-29b in muscle atrophy, we examined its expression level in myotubes differentiated from C2C12 or primary myoblasts treated with Dex. Notably, miR-29b was increased in both models (Fig. 2). In addition, we determined miR-29b expression level in two other in vitro models of muscle atrophy, including treatment of myotubes differentiated from C2C12 with TNF-α and $H_2O_2$, in which miR-29b was increased (Supplementary Fig. 2). Together, these data suggest that miR-29b is ubiquitously upregulated in muscle atrophy models both in vitro and in vivo, indicative of a potential functional role of miR-29b in this process.

**miR-29b controls muscle atrophy in vitro.** In fully differentiated C2C12 myotubes, miR-29b mimic was used to determine its role in promoting muscle atrophy. miR-29b mimic increased miR-29b expression level by 157-fold, without affecting miR-29a and miR-29c expressions (Fig. 3a), which confirms that the miR-29b mimic used in this study is specific to miR-29b. miR-29b overexpression reduced myotube diameter, elevated Atrogin-1 and Murf-1, decreased MHC and induced expressions of some autophagy-related genes (Map1-lc3b, Atg7, Atg12, Bnip3, Gabarapl1, Cathepsinl, Bnip3l and Vps34) and other ubiquitin ligases-related genes (Mul1, Traf6, Znf216, Cblb and Nedd4) (Fig. 3b–e). In addition, in myotubes differentiated from primary myoblasts, miR-29b overexpression also elevated Atrogin-1 and Murf-1, and decreased myotube diameter (Fig. 3f–h). To exclude the possibility that the above results achieved by miR-29b mimic might not be physiologically relevant, we also used miR-29b overexpression plasmid that increased miR-29b expression by 3.39-fold, and found that myotube diameter was reduced while Atrogin-1 and Murf-1 were elevated (Supplementary Fig. 3). Thus, miR-29b appears to be sufficient to promote muscle atrophy in vitro.

In fully differentiated C2C12 myotubes, miR-29b inhibitor was used to determine its role in regulating muscle size. miR-29b inhibitor decreased miR-29b expression level without affecting miR-29a and miR-29c expressions (Fig. 4a), which suggests that the miR-29b inhibitor used in this study is specific to miR-29b. Inhibition of miR-29b was not able to promote muscle hypertrophy in basal conditions, while it could abrogate the pro-atrophy effect of Dex stimulation (Fig. 4b,c). Similarly, miR-29b inhibitor also attenuated Dex-induced atrophy in myotubes differentiated from primary myoblasts (Fig. 4d). In addition, in myotubes differentiated from C2C12, treatment with TNF-α or $H_2O_2$ neither decreased myotube diameter or creatine kinase activity, nor induced the expression of Atrogin-1 and Murf-1, when miR-29b expression was inhibited (Supplementary Fig. 4). These results support the hypothesis that miR-29b is necessary for muscle atrophy.

Collectively, our findings illustrate that miR-29b is both necessary and sufficient for muscle atrophy in vitro.

**IGF-1 and PI3K(p85α) are target genes of miR-29b.** To investigate the mechanism by which miR-29b promotes muscle atrophy, we used the bioinformatic tool TargetScan to identify putative targets of miR-29b. Two potential targets identified were

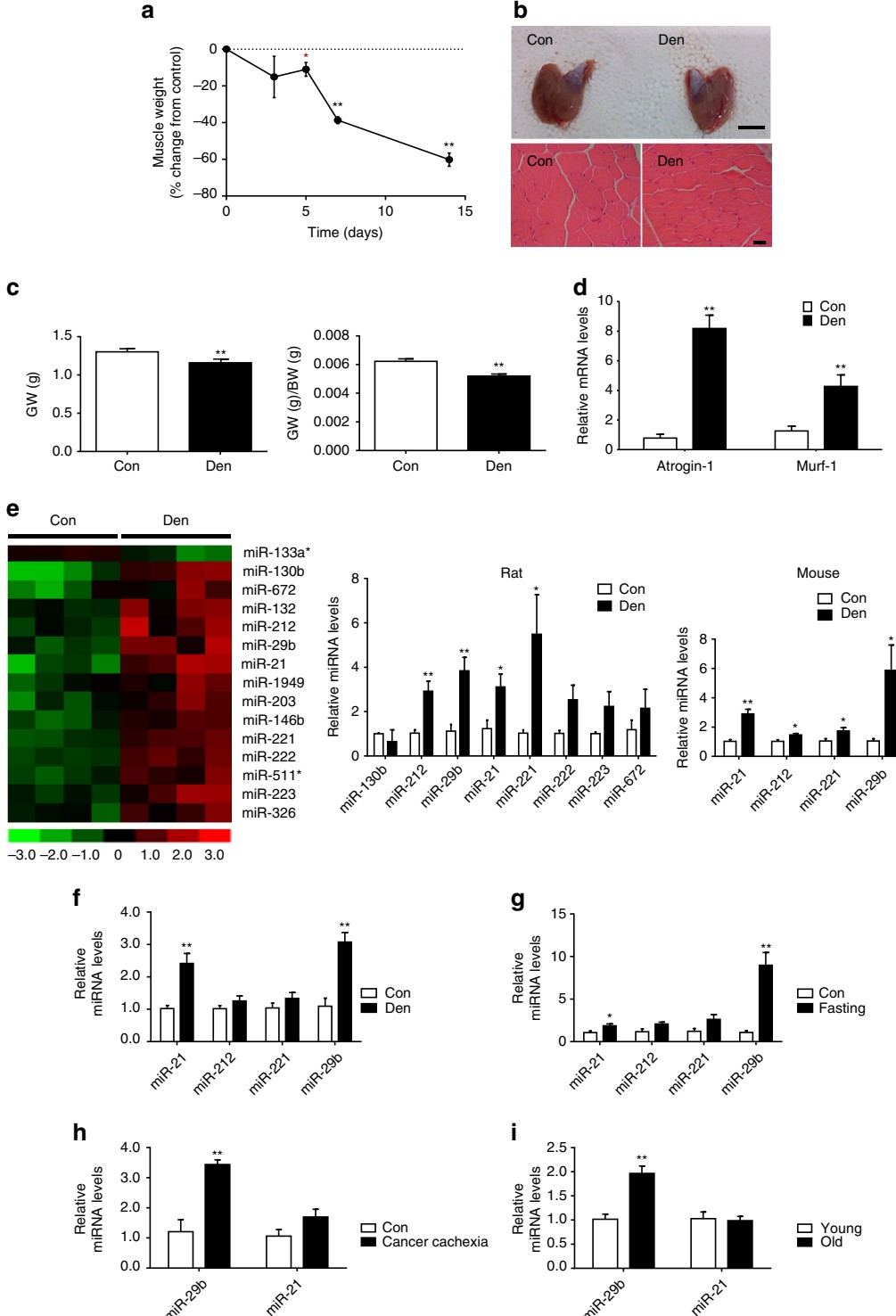

**Figure 1 | miR-29b is increased in multiple types of muscle atrophy *in vivo*.** (**a**) A time course of mass loss in the rat medial gastrocnemius muscle was examined in denervation model (*n* = 5 per group). (**b**) Denervation-induced marked muscle atrophy as determined by gastrocnemius muscle morphology (scale bar, 1 cm) and haematoxylin–eosin (HE) staining for muscle fibres (*n* = 5 per group, scale bar, 100 μm). (**c**) Gastrocnemius muscle weight (GW) and gastrocnemius muscle weight/body weight (GW/BW) ratio were both reduced in denervation rats (*n* = 5 per group). (**d**) qRT–PCR analysis showed increased *Atrogin-1* and *Murf-1* expressions in gastrocnemius muscle from denervation rats compared to controls (*n* = 5 per group). (**e**) miRNA arrays showed dysregulated miRNAs in gastrocnemius muscle from denervation rat model and qRT–PCR analysis of miRNA expressions in both rat and mouse models of denervation-induced muscle atrophy (*n* = 4 per group). (**f–i**) qRT–PCR analysis of miRNA expressions showed increased miR-29b in gastrocnemius muscle from dexamethasone (Dex)-, fasting-, cancer cachexia- and ageing-induced mouse muscle atrophy models (*n* = 5 for Dex, 5 for fasting, 5 for cancer cachexia and 4 for ageing). Con, Control. Den, Denervation. Error bars, s.e.m. An unpaired, two-tailed Student's *t*-test was used for comparisons between two groups. \**P* < 0.05, \*\**P* < 0.01.

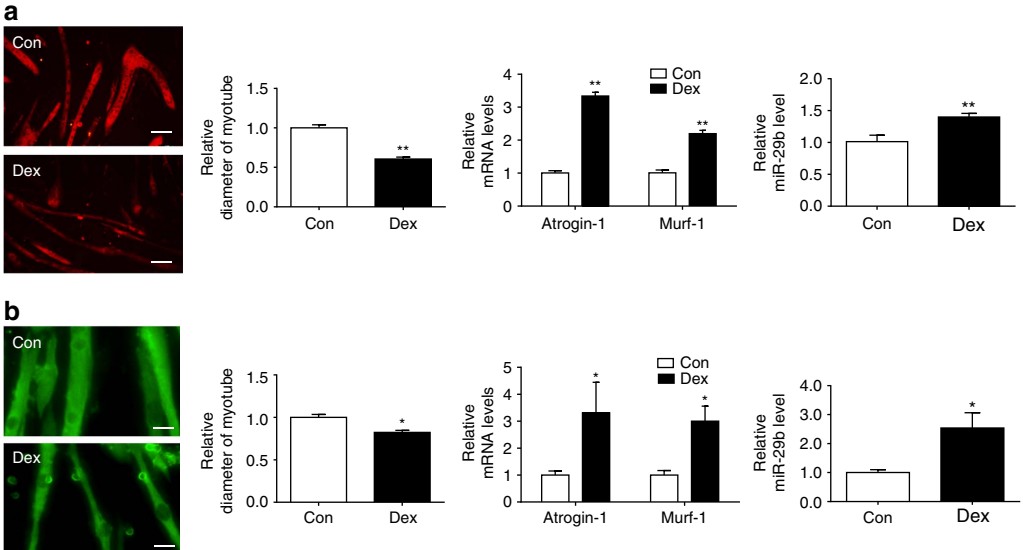

**Figure 2 | miR-29b is increased in multiple types of muscle atrophy *in vitro*.** (**a**) Immunofluorescent staining for C2C12 myotubes showed that Dex (50 μM)-induced muscle atrophy as evidenced by reduced myotube diameter, accompanied with increased *Atrogin-1*, *Murf-1* and miR-29b expressions (*n* = 4 per group, scale bar, 100 μm). (**b**) Immunofluorescent staining for myotubes differentiated from primary myoblasts showed that Dex (50 μM)-induced muscle atrophy as evidenced by reduced myotube diameter, accompanied with increased *Atrogin-1*, *Murf-1* and miR-29b expressions (*n* = 4 per group, scale bar, 100 μm). Dex, dexamethasone. Error bars, s.e.m. An unpaired, two-tailed Student's *t*-test was used for comparisons between two groups. *$P < 0.05$, **$P < 0.01$.

IGF-1 and PI3K(p85α), which are key proteins involved in the PI3K–AKT–mammalian target of rapamycin (mTOR) signalling pathway[25,26]. In non-dividing muscle cells, activation of this pathway stimulates protein synthesis and inhibits protein degradation. In atrophic cells, the PI3K–AKT–mTOR signalling is decreased[27].

We cloned the 3′untranslated region (UTR) of *IGF-1* and *PI3K(p85α)* in separate plasmids. Luciferase assays showed that exogenous miR-29b led to the reduction of luciferase activity in cells transfected with either the construct with 3′UTR of *IGF-1* or *PI3K(p85α)*, but had no effect when the putative miR-29b-binding site in either *IGF-1* or *PI3K(p85α)* 3′UTR was mutated (Fig. 5a). This suggests that IGF-1 and PI3K(p85α) are both direct targets of miR-29b.

The expressions of IGF-1 and PI3K(p85α) were decreased in differentiated C2C12 cells treated with Dex, TNF-α and H$_2$O$_2$: *in vitro* models of muscle atrophy (Supplementary Fig. 5a). In a parallel manner, transfection of miR-29b mimic into C2C12 myotubes resulted in decreased protein levels of IGF-1 and PI3K(p85α) (Fig. 5b). Conversely, transfection with miR-29b inhibitor resulted in increased expression of IGF-1 and PI3K(p85α) (Fig. 5b). These results suggest that miR-29b can regulate endogenous IGF-1 and PI3K(p85α) expression levels in skeletal muscle cells. Besides that, the downstream effectors (from IGF-1) were determined and we found that the phosphorylations of AKT (Ser-473), FOXO3A (Ser-253), mTOR and P70S6K were decreased by miR-29b mimic while all these phosphorylations were increased by miR-29b inhibitor, though the phosphorylations of AKT (Thr-308), FOXO3A (Thr-32) and 4EBP1 were not modulated (Fig. 5c,d).

To further assess if IGF-1 and PI3K(p85α) mediate the pro-atrophy effect of miR-29b, either IGF-1 or PI3K(p85α) overexpression plasmid was used to upregulate IGF-1 or PI3K(p85α) in the presence of miR-29b mimic. We found that IGF-1 or PI3K(p85α) overexpression could attenuate the pro-atrophy effect of miR-29b, as determined by myotube diameter and the expression levels of *Atrogin-1* and *Murf-1*

(Fig. 6). These results indicate that controlling IGF-1 and PI3K(p85α) expression is at least partly responsible for how miR-29b promotes muscle atrophy.

To determine whether these observations were also seen *in vivo*, we examined the expression levels of IGF-1, PI3K(p85α) and downstream effectors (from IGF-1) in our *in vivo* models. Consistent with the results above, IGF-1, PI3K(p85α) and the downstream effectors (from IGF-1) were also decreased in the gastrocnemius muscles from Den-, Dex- and fasting-induced atrophy models (Supplementary Figs 5b and 6).

**Yin Yang 1 triggers the upregulation of miR-29b.** To explore what triggers the upregulation of miR-29b, we firstly investigated whether a synergistic pathway that was controlled by the same IGF-1–AKT signalling was existent in a feed-forward loop to enhance protein degradation. Knockdown of *IGF-1* by siRNAs did not change the expression level of miR-29b (Fig. 7a), indicating that the synergistic pathway is unlikely existent. As Yin Yang 1 (YY1) has been reported to be the upstream of miR-29b in C2C12 myoblasts[28], we were interested in investigating if YY1 regulated miR-29b in C2C12 myotubes. We found that knockdown of *Yy1* by siRNAs increased miR-29b level in C2C12 myotubes (Fig. 7b). In addition, the expression level of YY1 was consistently downregulated in muscle atrophy induced by Den, Dex and fasting at both mRNA and protein levels (Fig. 7c,d). Importantly, in fully differentiated C2C12 myotubes, *Yy1* siRNAs reduced myotube diameter and elevated *Atrogin-1* and *Murf-1* expressions (Fig. 7e,f), suggesting its functional role in regulating muscle atrophy. Thus, YY1 might probably trigger the upregulation of miR-29b and contribute to muscle atrophy.

**miR-29b contributes to muscle atrophy *in vivo*.** To characterize the *in vivo* relevance of overexpressing miR-29b, we used miR-29b agomir to increase the expression level of miR-29b in mouse gastrocnemius muscles (Fig. 8 and Supplementary Fig. 7). Using this approach, we could increase miR-29b level by 2.5-fold

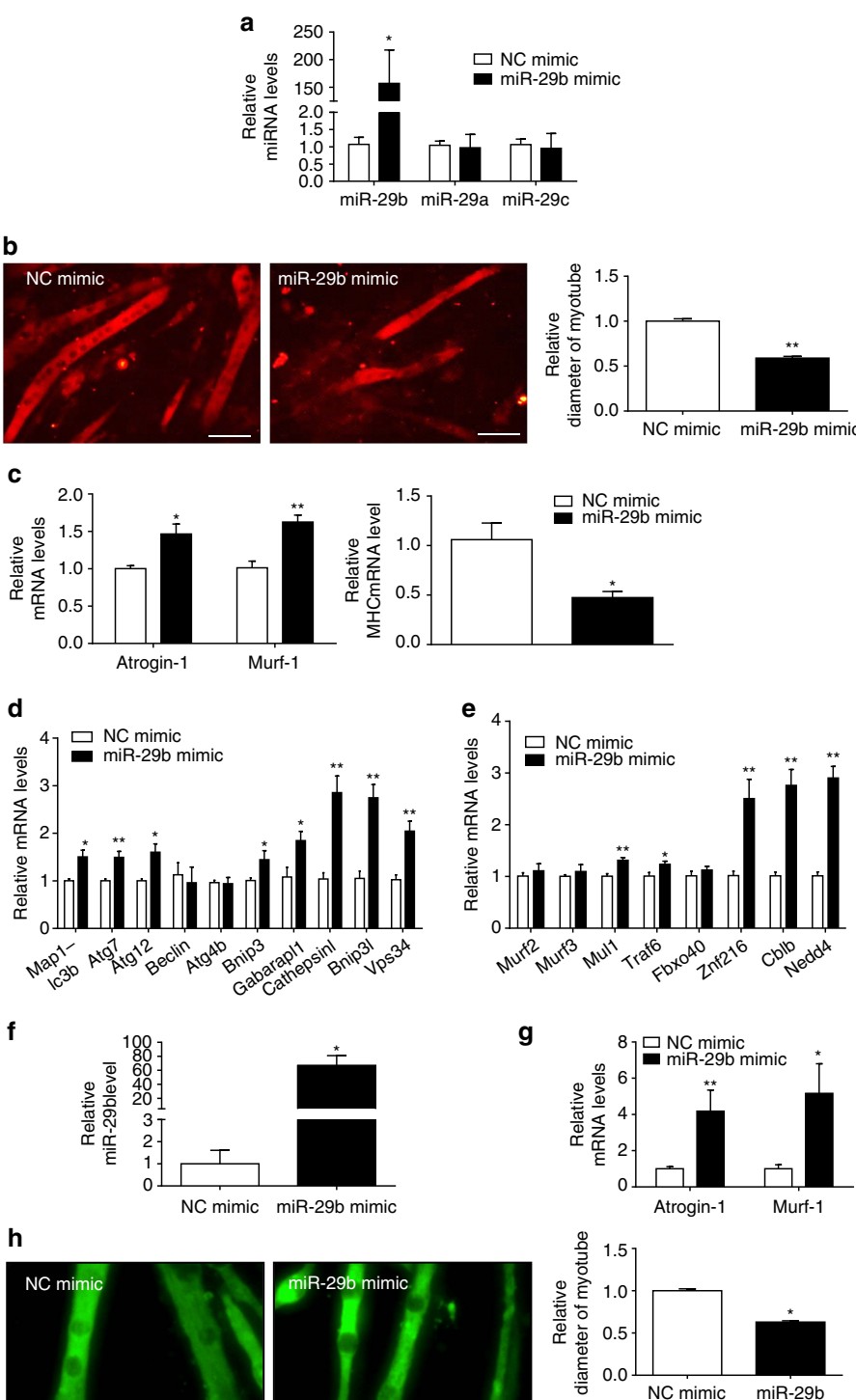

**Figure 3 | miR-29b is sufficient to induce muscle atrophy *in vitro*.** (**a**) qRT–PCR analysis showed increased miR-29b, but not miR-29a or miR-29c expressions, in C2C12 myotubes transfected with miR-29b mimic compared to negative control (NC mimic) ($n = 6$ per group). (**b**) Immunofluorescent staining for myotubes showed decreased myotube diameter in C2C12 myotubes transfected with miR-29b mimic ($n = 4$ per group, scale bar, 100 μm). (**c**) qRT–PCR analysis showed upregulated *Atrogin-1* and *Murf-1*, but downregulated *MHC* expressions in C2C12 myotubes transfected with miR-29b mimic ($n = 6$ per group). (**d**) qRT–PCR analysis showed increased autophagy-related gene expressions in C2C12 myotubes transfected with miR-29b mimic ($n = 6$ per group). (**e**) qRT–PCR analysis showed upregulation of other ubiquitin ligases-related gene expressions in C2C12 myotubes transfected with miR-29b mimic ($n = 6$ per group). (**f**) qRT–PCR analysis showed increased miR-29b expression in myotubes differentiated from primary myoblasts transfected with miR-29b mimic ($n = 6$ per group). (**g**) qRT–PCR analysis showed increased *Atrogin-1* and *Murf-1* expressions when myotubes differentiated from primary myoblasts were transfected with miR-29b mimic ($n = 6$ per group). (**h**) Immunofluorescent staining for myotubes differentiated from primary myoblasts showed that myotube diameter was decreased after transfected with miR-29b mimic ($n = 4$ per group, scale bar, 100 μm). Error bars, s.e.m. An unpaired, two-tailed Student's *t*-test was used for comparisons between two groups. *$P < 0.05$, **$P < 0.01$.

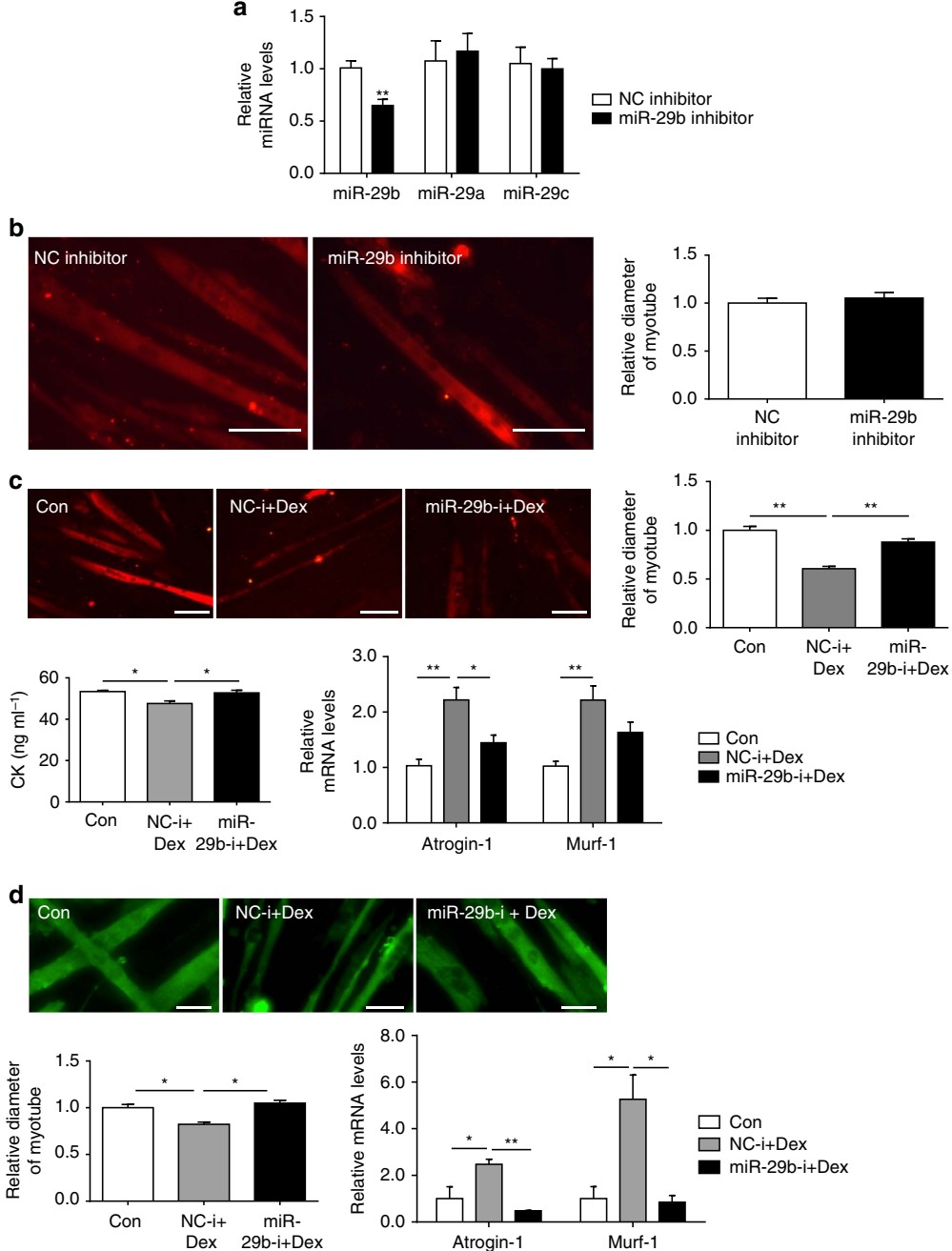

**Figure 4 | miR-29b is necessary for muscle atrophy *in vitro*.** (**a**) qRT–PCR analysis showed decreased miR-29b, but not miR-29a or miR-29c expressions, in C2C12 myotubes transfected with miR-29b inhibitor compared to negative control (NC inhibitor) ($n = 6$ per group). (**b**) Immunofluorescent staining for myotubes showed no difference in myotube diameter when C2C12 myotubes were transfected with NC inhibitor and miR-29b inhibitor ($n = 4$ per group, scale bar, 100 μm). (**c**) miR-29b inhibition abolished Dex (50 μM)-induced muscle atrophy in C2C12 myotubes, as determined by myotube diameter ($n = 4$ per group, scale bar, 100 μm), creatine kinase (CK) activity ($n = 6$ per group) and *Atrogin-1* and *Murf-1* mRNA levels ($n = 6$ per group). (**d**) miR-29b inhibition abolished Dex (50 μM)-induced muscle atrophy in myotubes differentiated from primary myoblasts, as determined by myotube diameter ($n = 4$ per group, scale bar, 100 μm), and *Atrogin-1* and *Murf-1* mRNA levels ($n = 6$ per group). Dex, dexamethasone. Error bars, s.e.m. An unpaired, two-tailed Student's *t*-test was used for comparisons between two groups (**a**,**b**). One-way ANOVA test was performed to compare multiple groups followed by Bonferroni's *post hoc* test (**c**,**d**). *$P < 0.05$, **$P < 0.01$.

without affecting miR-29a and miR-29c (Fig. 8a), with corresponding decrease in the targets as noted above (Supplementary Fig. 7h,i). Muscle atrophy was confirmed as evidenced by the decrease in gastrocnemius weight, gastrocnemius weight/body weight ratio, grip strength, myotube diameter (HE staining), mitochondria and glycogen content (periodic acid-schiff (PAS) and succinate dehydrogenase (SDH) stainings), MHC level and

mtDNA copy numbers (Fig. 8b–e,j,k,m); and the increase in some atrogenes including *Atrogin-1*, *Murf-1*, *Murf-2*, *Fbxo40*, *Traf6*, *Cblb* and *Nedd4* expressions, and protein ubiquitination and autophagy (Fig. 8h,i,l and Supplementary Fig. 7b,c). We further explored the atrophic fibre-type induced by miR-29b agomir, and consistently found that all types of fibres underwent atrophy as determined by SDH staining and qRT–PCR analysis of *Myh7*,

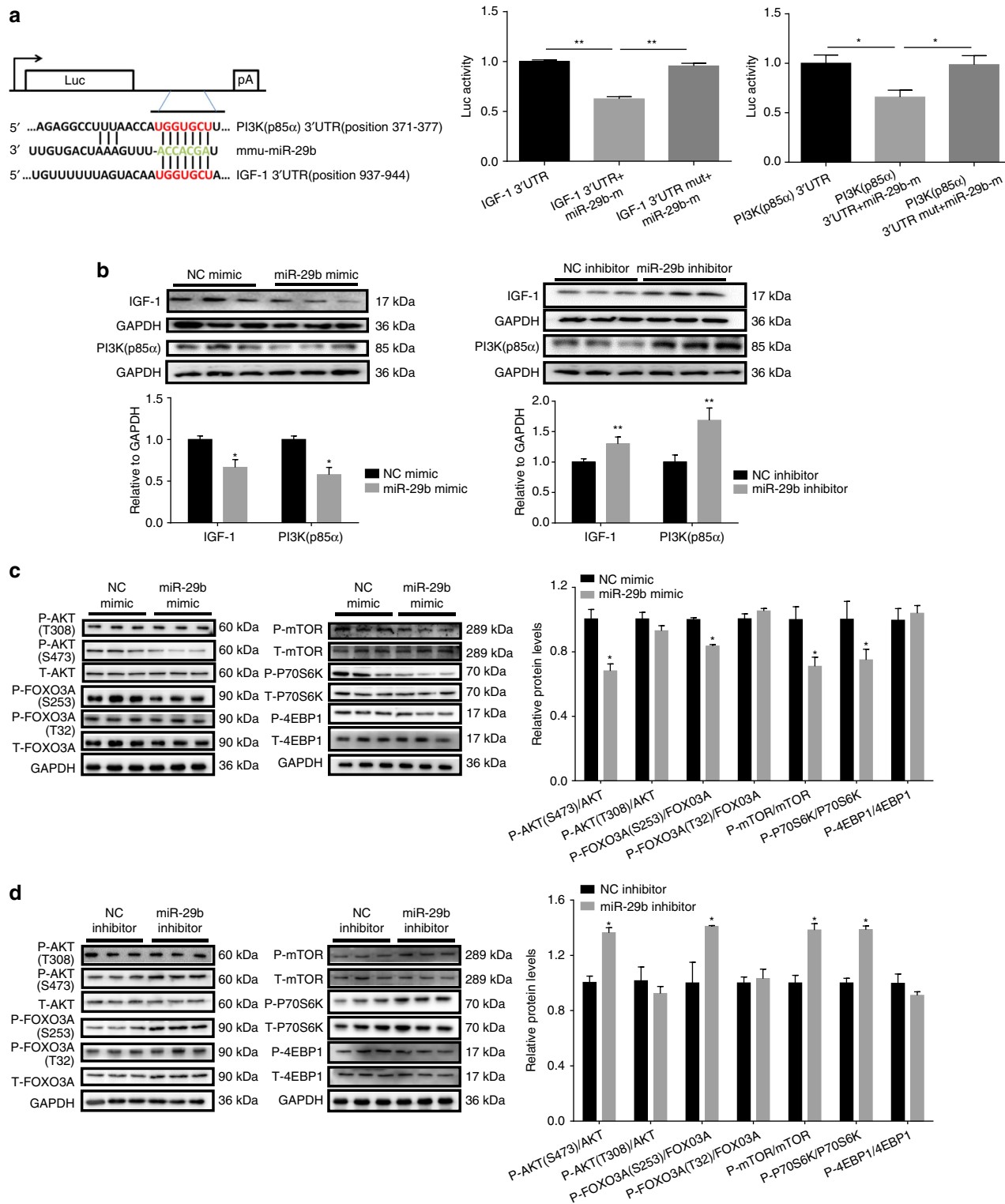

**Figure 5 | IGF-1 and PI3K(p85α) are identified as two target genes of miR-29b.** (**a**) TargetScan and Luciferase reporter assay showed that IGF-1 and PI3K(p85α) were two direct targets of miR-29b ($n = 5$ per group). (**b**) Western blot showed that miR-29b negatively regulated IGF-1 and PI3K(p85α) in C2C12 myotubes ($n = 3$ per group). (**c**) Western blot analysis for the AKT/FOXO3A/mTOR pathway (AKT, FOXO3A, mTOR, P70S6K, 4EBP1) showed decreased phosphorylation levels of AKT (S473), FOXO3A (S253), mTOR and P70S6K in C2C12 myotubes transfected with miR-29b mimic ($n = 3$ per group). (**d**) Western blot analysis for the AKT/FOXO3A/mTOR pathway (AKT, FOXO3A, mTOR, P70S6K, 4EBP1) showed increased phosphorylation levels of AKT (S473), FOXO3A (S253), mTOR and P70S6K in C2C12 myotubes transfected with miR-29b inhibitor ($n = 3$ per group). Error bars, s.e.m. The presented blots are representative samples of three independent experiments. An unpaired, two-tailed Student's $t$-test was used for comparisons between two groups (**b**–**d**). One-way ANOVA test was performed to compare multiple groups followed by Bonferroni's *post hoc* test (**a**). $*P < 0.05$, $**P < 0.01$.

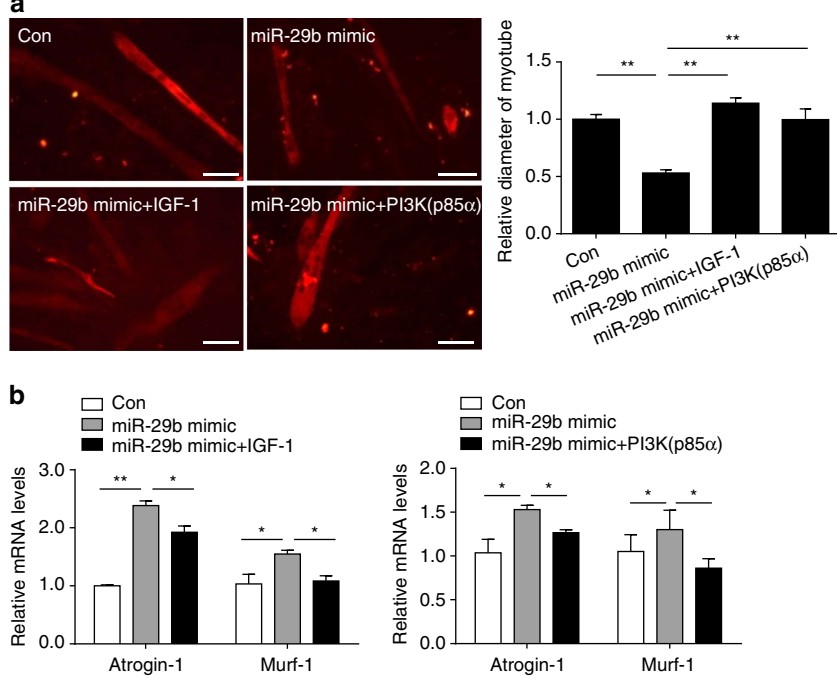

**Figure 6 | IGF-1 and PI3K(p85α) reduce miR-29b-induced muscle atrophy.** (**a**) Immunofluorescent staining for C2C12 myotubes showed that overexpression of IGF-1 or PI3K(p85α) reduced muscle atrophy induced by miR-29b mimic ($n = 4$ per group, scale bar, 100 µm). (**b**) qRT–PCR analysis showed that overexpression of IGF-1 or PI3K(p85a) reduced miR-29b-induced upregulation of *Atrogin-1* and *Murf-1* in C2C12 myotubes ($n = 6$ per group). Error bars, s.e.m. One-way ANOVA test was performed to compare multiple groups followed by Bonferroni's *post hoc* test. *$P < 0.05$, **$P < 0.01$.

*Myh2*, *Myh4*, *Myh1* encoding myosin isoforms MyHC-I, MyHC-IIa, MyHC-IIb and MyHC-IIx/d (refs 29,30) (Fig. 8f,g). In addition, the proportion of type I fibre in fibre-type composition was decreased while that of type IIB fibre was increased as determined by immunofluorescent staining with MHC antibodies (Supplementary Fig. 7a). However, no obvious inflammation or degeneration/ regeneration events occurred as determined by qRT–PCRs for *Myod*, *Myog*, *Myf5*, *IL-1α*, *IL-1ß* and *IL-6* (Supplementary Fig. 7d,e). Besides that, although denervation significantly increased denervation markers including *Musk*, *Achra*, *Achre*, *Achrg*, *Cpla2*, *Ncam* and *Runx1*, miR-29b agomir only slightly elevated *Achre* and *Cpla2* while other denervation markers were at largely unaffected (Supplementary Fig. 7f,g). Interestingly, in mice injected with miR-29b agomir, expression of IGF-1, PI3K(p85α) and the downstream (from IGF-1) effectors in gastrocnemius muscles were decreased compared to control (Supplementary Fig. 7h,i). Thus, these data indicate that the increase of miR-29 is able to induce muscle atrophy *in vivo*.

To investigate whether inhibiting miR-29b attenuates muscle atrophy, we treated mice with intramuscular injection of miR-29b sponge, followed by denervation of the right sciatic nerve. miR-29b sponge was able to decrease miR-29b expression level in the gastrocnemius without affecting miR-29a and miR-29c expressions (Fig. 9a). All mice were euthanized 5 days later and miR-29b sponge significantly decreased miR-29b level (Fig. 9b). In the absence of miR-29b sponge, denervation decreased the ratio of gastrocnemius weight to body weight by 20.6%. In comparison, miR-29b sponge injections led to a 44.8% reduction in denervation-induced muscle atrophy as determined by the ratio of gastrocnemius weight to body weight (Fig. 9c,d). Similarly, the decrease in gastrocnemius weight, diameter of muscle fibres and increase in *Atrogin-1* and *Murf-1* expression levels in denervated muscles were also attenuated in mice injected with miR-29b sponge (Fig. 9c,e,f). Interestingly, in denervation

mice injected with miR-29b sponge, expression levels of IGF-1, PI3K(p85α) and the downstream (from IGF-1) effectors in gastrocnemius muscles were increased compared to control (Supplementary Fig. 8).

As immobilization of limbs is a common clinical procedure for orthopaedic medicine, we also explored the role of miR-29b inhibition in muscle atrophy induced by immobilization of limbs. Muscle atrophy was induced by immobilization of limbs as evidenced by decreased gastrocnemius weight and gastrocnemius weight/body weight ratio, and elevated *Atrogin-1* and *Murf-1*, accompanied with an increase of miR-29b (Supplementary Fig. 9a–c). Besides that, we also confirmed that the upregulation of miR-29b in immobilization-induced muscle atrophy was a generalized process as it was consistently increased in TA, soleus, and EDL (Supplementary Fig. 9d). A single intramuscular injection of miR-29b sponge in gastrocnemius muscle significantly inhibited miR-29b expression (Supplementary Fig. 9e). Muscle atrophy was largely attenuated as evidenced by increased gastrocnemius weight and gastrocnemius weight/body weight ratio, decreased *Atrogin-1* and *Murf-1*, and increased muscle fibre diameter, accompanied by upregulated expressions of IGF-1 and PI3K(p85α) (Supplementary Fig. 9f–i). These data demonstrate that suppression of miR-29b has anti-atrophy effect and could at least partly attenuate muscle atrophy.

## Discussion

Muscle atrophy can be commonly induced by a variety of stress and is debilitating[31,32]. The pathogenesis of skeletal muscle atrophy is complex and remains incompletely understood[31,33]. Muscle atrophy in a variety of conditions such as cancer, denervation, disuse and fasting shares a common mechanism in the induction of *Atrogin-1* and *Murf-1*. Interestingly, in aged skeletal muscle, *Atrogin-1* and *Murf-1* have been reported

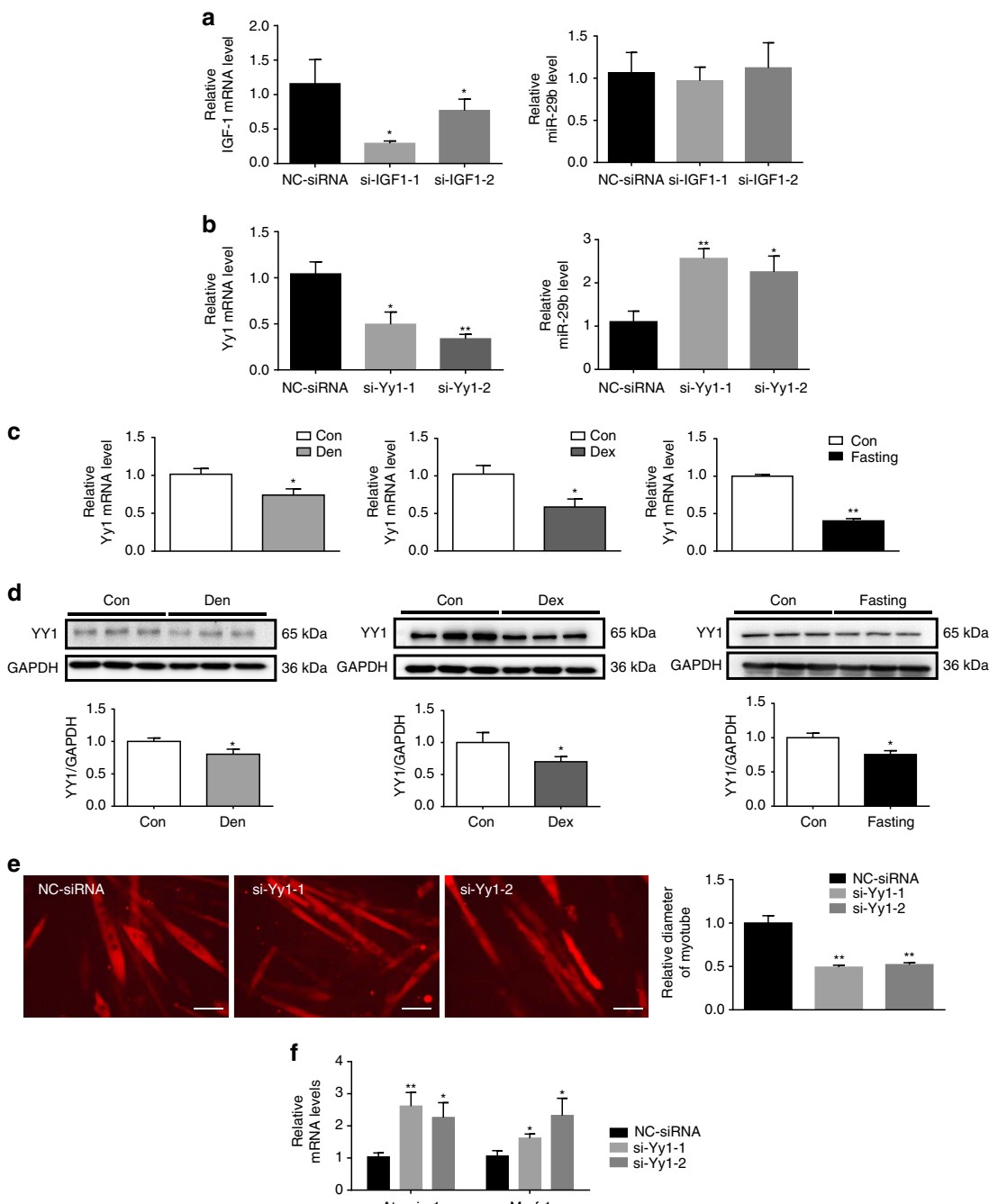

**Figure 7 | YY1 negatively regulates miR-29b in muscle atrophy.** (**a**) qRT–PCR analysis showed downregulated *IGF-1* but not miR-29b expression in C2C12 myotubes transfected with *IGF-1* siRNAs compared to negative control (NC siRNA) ($n = 6$ per group). (**b**) qRT–PCR analysis showed down-regulated *Yy1* while upregulated miR-29b expressions in C2C12 myotubes transfected with *Yy1* siRNAs compared to negative control (NC siRNA) ($n = 6$ per group). (**c**) qRT–PCR analysis showed decreased *Yy1* mRNA level in the gastrocnemius from Den-, Dex- and fasting-induced muscle atrophy models ($n = 5$ per group). (**d**) Western blot analysis showed decreased YY1 protein level in the gastrocnemius from Den-, Dex- and fasting-induced muscle atrophy models ($n = 3$ per group). (**e**) Immunofluorescent staining for C2C12 myotubes showed decreased myotube diameter when transfected with *Yy1* siRNAs ($n = 4$ per group, scale bar, 100 μm). (**f**) qRT–PCR analysis showed increased *Atrogin-1* and *Murf-1* expressions when C2C12 myotubes were transfected with *Yy1* siRNAs ($n = 6$ per group). Den, Denervation; Dex, dexamethasone. Error bars, s.e.m. The presented blots are representative samples of three independent experiments. An unpaired, two-tailed Student's *t*-test was used for comparisons between two groups (**c,d**). One-way ANOVA test was performed to compare multiple groups followed by Bonferroni's *post hoc* test (**a,b,e,f**). $*P < 0.05$, $**P < 0.01$.

increased, unchanged or decreased, and these discrepancies might be caused by difference in the muscles, species and strains and genders[34–38]. We here showed that in aged male mice, *Atrogin-1* and *Murf-1* were increased in gastrocnemius muscles. Aberrant expression of miRNAs has been reported in muscle atrophy[39].

Despite that investigators have demonstrated the presence of common atrophy genes that are coordinately regulated in several models of atrophy, few studies have examined the role of miRNAs that were ubiquitously altered, and perhaps played a central role in models of atrophy[1]. Here we report that miR-29b

is commonly upregulated in multiple types of muscle atrophy. Interestingly, a study has reported that the soleus and the gastrocnemius muscle had contrasting regulation of miR-1, which was decreased in the soleus muscle 1 week post crush injury and nerve transection, while its expression was significantly increased in the gastrocnemius muscle[40]. We thus investigated if miR-29b upregulation in denervation was specific in gastrocnemius muscles or it was a more generalized process, and we found that miR-29b was consistently elevated in all tested muscles including gastrocnemius, TA, soleus and EDL, further demonstrating that miR-29b is a common target for muscle atrophy. As miRNAs are emerging as promising therapeutic candidates for drug development[16], our results identify a novel target in this important clinical space.

miRNAs participate in multiple regulatory pathways in skeletal muscle[16]. A cluster of myomiRs including miR-1, miR-133 and miR-206 have been found to play important roles in regulating myogenesis and muscle regeneration[16]. Many profiling

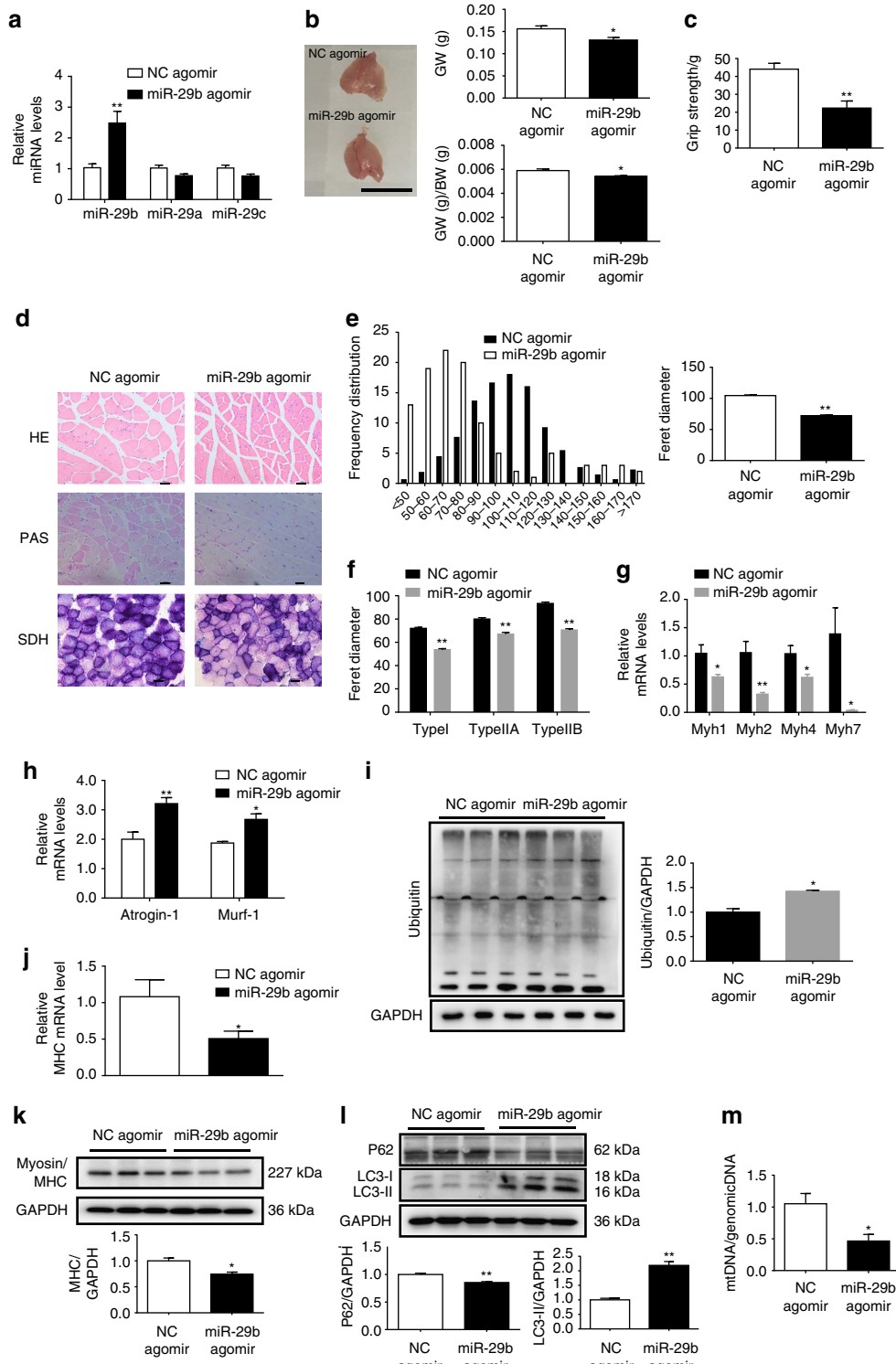

experiments have been performed in muscle atrophy. Fourteen miRNAs have been identified to be dysregulated in skeletal muscle from old men comparing to that from young men[41]. Another study has revealed that 75 miRNAs were differently expressed between the young and old men groups (40 upregulated and 35 downregulated)[42]. In addition, accumulating evidence suggests that the aberrant expressions of miRNAs, such as miR-1, miR-133, miR-23a, miR-206, miR-27, miR-628, miR-431 and miR-21 (refs 17–24), contribute to muscle atrophy. Most of these miRNAs have been reported to be upregulated in individual model of muscle atrophy. Akin to the dysregulation of atrophy-associated proteins such as the ubiquitin ligases called *Atrogin-1* and *Murf-1* that are altered in most models of atrophy[39], miRNAs universally dysregulated in diverse atrophy models have not yet been identified[20]. Previously, miRNA profiling of mouse muscles under several wasting conditions including denervation, fasting, diabetes and cancer cachexia has been performed and a peculiar pattern of miRNA expressions for each catabolic condition has been identified[20]. Here we used a different strategy namely profiling miRNAs in denervated muscle and validated them using qRT–PCRs in several other muscle atrophy models. We provide direct evidence that miR-29b is commonly upregulated in five different models of skeletal muscle atrophy, including muscle atrophy induced by denervation, Dex, fasting, ageing and cancer cachexia, indicating that upregulation of miR-29b might be a common driver of muscle atrophy.

miR-29 has been reported to function as a positive regulator of myogenesis through feedback inhibition of the transcription factor YY1 (ref. 28). Similarly, decreased miR-29 was reported to suppress myogenesis in chronic kidney disease by targeting YY1 (ref. 43). miR-29 has also been demonstrated to impair muscle progenitor cell proliferation, increase cell cycle arrest protein levels, and induce cellular senescence in ageing muscle by targeting PI3K(p85α), IGF-1 and B-myb[44]. Here we observed that miR-29b was necessary and sufficient to promote muscle atrophy in *in vitro* models of muscle atrophy, including myotubes differentiated from C2C12 treated with Dex, TNF-α and $H_2O_2$ and from primary myoblasts treated with Dex. Most importantly, miR-29b was able to induce muscle atrophy *in vivo* while inhibition of miR-29b attenuated denervation- and immobilization-induced muscle atrophy. It is interesting that miR-29b can induce senescence and atrophy. However, the effects of miR-29b in inducing cellular senescence are explored in muscle progenitor cell while its effect in inducing atrophy is investigated in myotubes, respectively, supporting the cellular-specific effects of miR-29b (ref. 44). Of note, miR-29b decreased phosphorylation of FOXO3A at serine-253 and thus induced *Atrogin-1* and *Murf-1*

expressions, leading to muscle atrophy. Interestingly, we showed that the proportion of type I fibre in fibre-type composition was decreased while that of type IIB fibre was increased, and also observed a decrease in strength phenotype. We think this result could be explained as follows. First, all types of fibres underwent atrophy. Second, qRT–PCR analysis for *Myh7*, *Myh2*, *Myh4*, *Myh1* encoding myosin isoforms MyHC-I, MyHC-IIa, MyHC-IIb and MyHC-IIx/d showed that all myosin isoforms were decreased and the decrease of *Myh7* was most significant, making the relative increase of type IIB fibre. Collectively, our data suggest that miR-29b is sufficient and necessary for multiple types of muscle atrophy.

The IGF-1–PI3K–AKT signalling is critical for controlling the balance between protein synthesis and degradation[45,46]. Deactivation of this signalling will result in decreased protein synthesis and increased protein degradation, which may lead to muscle atrophy[8,46,47]. Based on bioinformatics analysis and further experimental validation, IGF-1 and PI3K(p85α) were identified as two target genes of miR-29b in myotubes. IGF-1 has previously been recognized as a critical factor for coordinating muscle growth and increasing muscle mass[7,8]. IGF-1 has been reported to inversely regulate atrophy-induced genes via the PI3K/AKT/mTOR pathway, and the IGF-1/PI3K/AKT pathway can prevent expressions of muscle atrophy-induced ubiquitin ligases by inhibiting FOXO transcription factors[7,8]. PI3K(p85α) is the regulatory subunit of PI3K and the loss of class IA PI3K signalling in muscle has been reported to induce impaired muscle growth[48]. Our functional experiments in myotubes confirmed that suppression of IGF-1 and PI3K(p85α) was responsible for the pro-atrophy effect of miR-29b in myotubes. In addition, we found that the phosphorylations of AKT (Ser-473), FOXO3A (Ser-253), mTOR and P70S6K were decreased by miR-29b mimic while all these phosphorylations were increased by miR-29b inhibitor. Importantly, we found that IGF-1, PI3K(p85α) and the downstream (from IGF-1) effectors were all decreased in the *in vitro* muscle atrophy models and miR-29b agomir could decrease them *in vivo*, providing some insights of their potential roles in muscle atrophy *in vivo*. Nevertheless, it would be highly interesting to investigate *in vivo* therapeutic roles for miR-29b targets individually or together based on gain-of-function and loss-of-function experiments. Moreover, the other relevant targets of miR-29b in regulating atrogenes, fibre type and autophagy pathways should also be identified in the future.

In conclusion, miR-29b contributes to multiple types of muscle atrophy via targeting of IGF-1 and PI3K(p85α), and that suppression of miR-29b may represent a therapeutic approach for muscle atrophy induced by different stimuli.

**Figure 8 | miR-29b is sufficient to induce muscle atrophy *in vivo*.** (**a**) qRT–PCR analysis showed increased miR-29b, but not miR-29a or miR-29c expressions, in mice treated with miR-29b agomir compared to negative control (NC agomir) (n = 5 per group). (**b**) miR-29b agomir induced muscle atrophy, as determined by gastrocnemius muscle morphology, gastrocnemius weight (GW) and gastrocnemius weight/body weight (GW/BW) ratio (n = 5 per group, scale bar, 1 cm). (**c**) The grip strength of right hind limb was reduced in miR-29b agomir-treated mice (n = 5 per group). (**d**) miR-29b agomir-induced muscle atrophy was also evidenced by haematoxylin–eosin (HE) staining, periodic acid-schiff (PAS) staining and succinate dehydrogenase (SDH) staining (n = 5 per group, scale bar, 50 μm). (**e**) Quantification of muscle fibre diameter distribution confirmed that miR-29b agomir induced muscle atrophy (n = 5 per group). (**f**) Quantification of diameter of different myofibre types showed that all types of fibres underwent atrophy in miR-29b agomir-treated mice (n = 5 per group). (**g**) qRT–PCR analysis showed decreased *Myh1*, *Myh2*, *Myh4* and *Myh7* expressions in mice treated with miR-29b agomir (n = 5 per group). (**h**) qRT–PCR analysis showed upregulated *Atrogin-1* and *Murf-1* expressions in mice treated with miR-29b agomir (n = 5 per group). (**i**) Western blot analysis showed upregulation of Ubiquitin protein expressions in mice treated with miR-29b agomir (n = 3 per group). (**j**) qRT–PCR analysis showed downregulated *MHC* in mice treated with miR-29b agomir (n = 5 per group). (**k**) Western blot analysis showed reduced MHC protein level in mice treated with miR-29b agomir (n = 3 per group). (**l**) Western blot analysis showed downregulated P62 but up-regulated LC3-II protein levels in mice treated with miR-29b agomir (n = 3 per group). (**m**) qRT–PCR analysis showed that mtDNA copy number was decreased in miR-29b agomir-treated mice (n = 5 per group). Age- and sex-matched mice were used for experiments randomly. Error bars, s.e.m. An unpaired, two-tailed Student's *t*-test was used for comparisons between two groups. *P < 0.05, **P < 0.01.

## Methods

**Animal experiments.** All animals were purchased from Shanghai Institutes for Life Science of the Chinese Academy of Sciences (Shanghai, China). The standard pellet diet and water were provided *ad libitum* and all animals were maintained on a 12 h light/12 h dark cycle in a temperature-controlled room at 21–23 °C. All procedures with animals were in accordance with the guidelines on the use and care of laboratory animals for biomedical research published by National Institutes of Health (No. 85-23, revised 1996), and the experimental protocol was reviewed

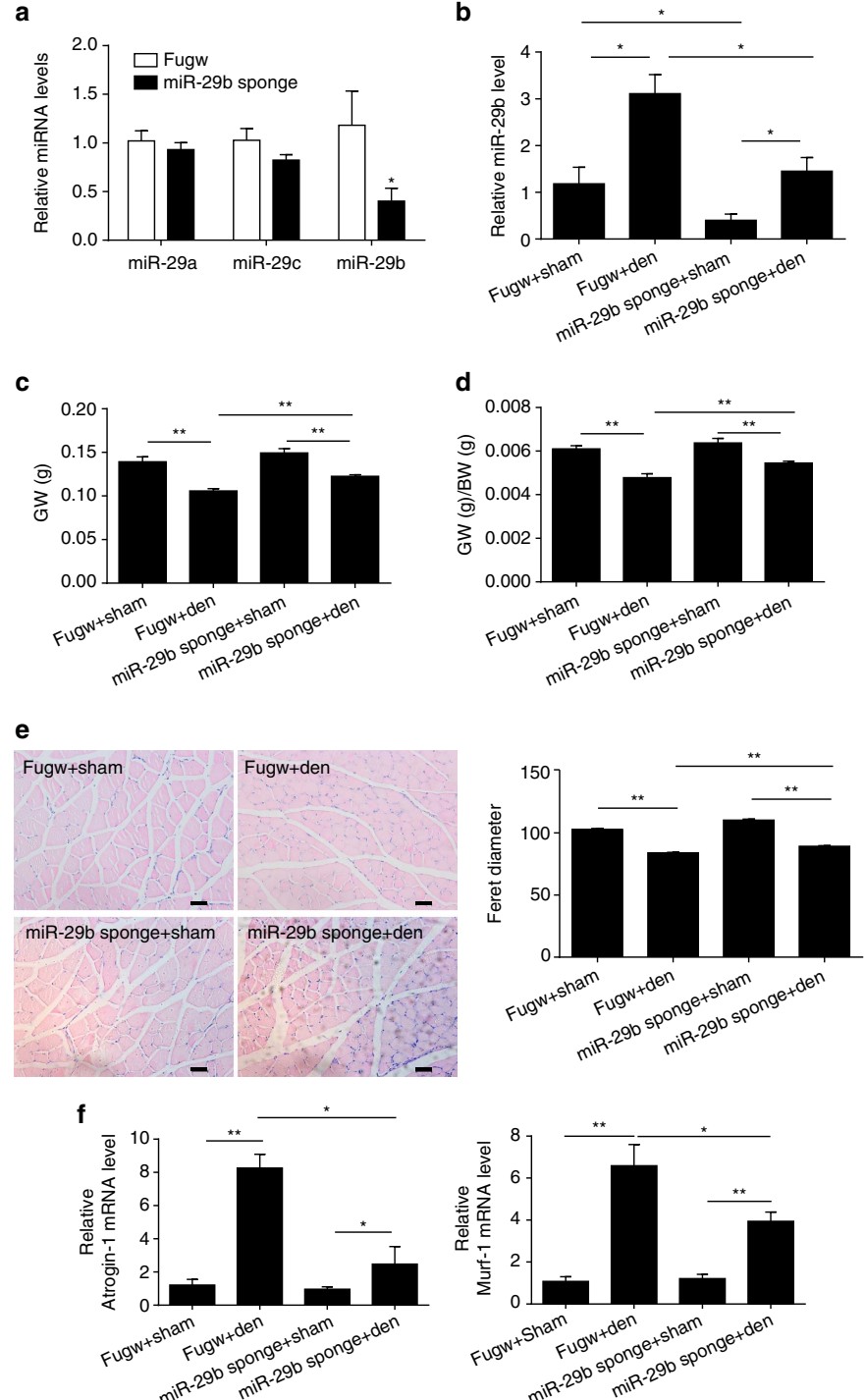

**Figure 9 | miR-29b is necessary for muscle atrophy *in vivo*.** (**a**) qRT–PCR analysis showed reduced miR-29b, but not miR-29a or miR-29c expressions, in mice treated with miR-29b sponge compared to fugw control ($n = 5$ per group). (**b**) qRT–PCR analysis showed reduced miR-29b expression level in miR-29b sponge-treated mice in the presence or absence of denervation (Den) ($n = 5$ per group). (**c,d**) Gastrocnemius weight (GW) and gastrocnemius weight/body weight (GW/BW) ratio showed that miR-29b sponge at least partly blocked denervation-induced muscle atrophy ($n = 5$ per group). (**e**) Haematoxylin–eosin (HE) staining demonstrated increased muscle fibre diameter in denervated mice treated with miR-29b sponge compared to those treated with fugw control ($n = 5$ per group, scale bar, 50 µm). (**f**) qRT–PCR analysis showed downregulated *Atrogin-1* and *Murf-1* expressions in denervated mice treated with miR-29b sponge compared to those treated with fugw control ($n = 5$ per group). Age- and sex-matched mice were used for experiments randomly. Error bars, s.e.m. An unpaired, two-tailed Student's *t*-test was used for comparisons between two groups (**a**). One-way ANOVA test was performed to compare multiple groups followed by Bonferroni's *post hoc* test (**b**–**f**). *$P < 0.05$, **$P < 0.01$.

and approved by the ethical committees of School of Life Science, Shanghai University.

**Muscle atrophy models.** Eight-week-old male C57BL/6 or BALB/c mice and 7–8-week-old male Sprague Dawley rats were used in this study. Multiple muscle atrophy models were established as follows: (1) the denervation-induced muscle atrophy was generated in rats or C57BL/6 mice by cutting off the mid-thigh region of the right sciatic nerve. The sham was generated by the same process but without cutting off sciatic nerve. Rats or mice were killed at 3, 5, 7 and 14 days after denervation, respectively. (2) The Dex-induced muscle atrophy model was induced by treating C57BL/6 mice with either Dex or phosphate-buffered saline (PBS; control) via intraperitoneal injections at a dose of $25\,mg\,kg^{-1}$ per day. All mice were killed after 1 week. (3) For fasting-induced muscle atrophy, C57BL/6 mice were maintained for 48 h with no food but free access to water, and the control mice were fed normally. (4) For ageing-induced muscle atrophy model, 10-week-old C57BL/6 mice were served as control and 23-month-old mice were collected as the ageing group. (5) For immobilization-induced muscle atrophy, the right ankle joint of C57BL/6 mice was fixed at 90° of flexion by insertion of a screw ($0.4 \times 8\,mm$) through the calcaneus and talus into the shaft of the tibia[1]. All mice were killed after 1 week. (6) For cancer cachexia-induced muscle atrophy model, BALB/c mice were subcutaneously inoculated with $10^6$ of mouse colon cancer C26 cells. All mice were killed after 2 weeks.

**miRNA arrays.** Total RNA extracted from the gastrocnemius muscles from denervation-induced muscle atrophy rat model was used for miRNA arrays. miRNA profiling was performed with OE Biotech's (Shanghai, China) miRNA microarray service based on Affymetrix miRNA 3.0 Array. The MIAME-compliant data have been submitted to Gene Expression Omnibus (GEO, platform ID: GSE81914).

**Cell culture and transfection.** C2C12 cells (mouse skeletal myoblasts) were obtained from ATCC and were tested and found negative for mycoplasma contamination before use. C2C12 cells were cultured in Dulbecco's modified Eagle's medium (DMEM) containing $4.5\,g\,l^{-1}$ glucose with 10% fetal bovine serum at 37 °C with 5% $CO_2$. To induce differentiation, cells were planted on culture plates coated with 0.1% gelatin, and when cell confluence reached 70%, the medium was switched into differentiation medium (DMEM containing 2% horse serum). After 4 days, multinuclear myotubes were formed.

To induce different types of muscle atrophy in vitro, cells were incubated with 50 μM Dex (or $100\,ng\,ml^{-1}$ TNF-α or 400 μM $H_2O_2$) in 2% horse serum in DMEM for 24 h. After incubation, cells were harvested or used for morphological analysis.

Myotube transfection was performed with Lipofectamine2000 Reagent (Invitrogen) according to the manufacturer's instructions. The mimic negative control, miR-29b mimic, inhibitor negative control and miR-29b inhibitor were bought from RiboBio. The transfection dosages of mimic and inhibitor were 50 and 100 nM, respectively. The transfection dosage of siRNAs for IGF-1 and Yy1 was 100 nM and the sequence was listed in Supplementary Table 2. The transfection was performed after myotubes formed and 24 h later, Dex (or TNF-α or $H_2O_2$) was added and cells were further cultured for 24 h.

**Primary myoblasts isolation and differentiation.** Primary myoblasts were isolated from 3-week-old C57BL/6 mice. Briefly, hind limb muscles were harvested, finely minced and then digested in collagenase solution. Cell resuspension was pre-plated for 30 min to remove fibroblasts. Unattached cells were collected in the growth medium for further application. To culture and induce differentiation, the isolated primary myoblasts were differentiated for four days with 5% horse serum solution on collagen-coated plates.

**Luciferase reporter assays.** The 3′UTRs of IGF-1 and PI3K (p85α) gene were amplified by PCR and inserted into the firefly luciferase reporter PGL3-basic Vector (Promega). The primers used were as follows: IGF-1 3′UTR, forward: 5′-<u>TCTAGA</u>ACAA**GGTGCTA**TTTTGTAGTT TG-3′, reverse: 5′-<u>TCTAGA</u>GG AGGGCCTTTGCATCTCCC-3′; PI3K (p85α) 3′UTR, forward: 5′-<u>TCTAGA</u>ACC A**TGGTGCT**TGTTAACGC-3′, reverse: 5′-<u>TCTAGAC</u> ACCCAG**GCTACA** CCAG-3′ (the bold part is the seed sequences). Mutation in the miR-29b target site was generated by PCR from the plasmid PGL3-3′UTR of IGF-1 or PI3K (p85α). The primer used were as follows: IGF-1 3′UTR mutation, forward: 5′-<u>TCTAGA</u>A CAA**ACCACGAT**TTTTGTAGTTTG-3′, reverse: 5′-<u>TCTAGA</u>GGA GGCCTTT GCATCTCCC-3′; PI3K (p85α) 3′UTR mutation, forward: 5′-<u>TCTAGA</u>ACCA **ACCACGA**TGTTAACGC-3′, reverse: 5′-<u>TCTAGA</u>CACCCA<u>GG CTACACC</u> AG-3′ (the bold part is the mutation seed sequences) and the underlined sequences are the digestion sequences of XbaI.

HEK293T were co-transfected with 200 ng PGL3-basic-3′UTR (or 3′UTR mut), 5 ng Renila (used as an internal control) and 50 nM miR-29b mimic or mimic negative control using Lipofectamine2000 Reagent in 24-well plates for 48 h. The activation of firefly and Renila luciferase was analysed by a dual-luciferase reporter assay Kit (Promega) according to the manufacturer's instructions.

**Plasmids.** PI3K (P85α): pBS-p85α was a gift from Lewis Cantley (Addgene plasmid # 1407). The sequence of an overexpressed IGF-1 was obtained from the NCBI, and the gene fragment was obtained by PCR. The primers used were as follows: forward 5′-GGGGAATTCATGACCGCACCTGCAATAAAG-3′ and reverse 5′-GGGTCTAGACTAGCCCAGTCTTTTTTCTCTG-3′. The CDS sequences of IGF-1 were ligated into pEGFP-C3. The sequence of pre-miR-29b was obtained from the NCBI, and the gene fragment was obtained by PCR. The primers used were as follows: forward 5′-GGGGGATCCACTTACTTCAGGGCTGT ACACTCA-3′ and reverse 5′-GGGC TCGAGAGGTCAGCATAGGATCGC CTG-3′. The sequences of pre-miR-29b were ligated into Fugw.

**Quantitative real-time polymerase chain reactions.** Total RNA extraction from muscles and cells was performed by RNeasy Mini Kit (Qiagen), according to the manufacturer's instructions. The Bulge-Loop miRNA qPCR Primer Set (RiboBio) was used to determine the expression levels of miRNAs by qRT–PCRs with Takara SYBR Premix Ex Taq (TliRNaseH Plus) in a BioRad CFX96 Real-Time PCR Detection System. 5S was used as an internal control. For mRNA analysis, cDNA was synthesized using Takara PrimeScript 1st Strand cDNA Synthesis Kit and was subjected to quantitative PCR with Takara SYBR Premix Ex TaqTM. 18S was used as an internal control. The primer sequences used in this study were listed in Supplementary Table 3. The relative expression level of gene or miRNA was calculated using the $2^{-\Delta\Delta Ct}$ method.

**Western blot.** Protein samples were extracted from muscles or cells by using RIPA buffer (KeyGEN, China) with a protease inhibitor cocktail (KeyGEN, China). The concentration of protein sample was determined by the BCA Protein Assay Kit (TaKaRa). Equal amounts of protein samples were separated by SDS–polyacrylamide gel electrophoresis gel electrophoresis, and then were transferred to polyvinylidene difluoride membrane. After that, the membranes were blocked with 5% bovine serum albumin (BSA) for 1 h at room temperature. Primary antibodies were incubated and a horseradish peroxidase-conjugated secondary antibody was followed. The primary antibodies used were as follows: IGF-1 (1:1,000, Bioworld Technology, Inc.), PI3K (p85α) (1:500, Cell Signaling Technology, Inc.), YY1 (1:1,000, Proteintech, Inc.), FOXO3A (1:1000, Abclonal Technology, Inc.), P-AKT (T308) (1:1,000, Cell Signaling Technology, Inc.), P-AKT (S473) (1:1,000, Cell Signaling Technology, Inc.), AKT (1:1,000, Proteintech, Inc.), P-FOXO3A (S253) (1:1,000, Cell Signaling Technology, Inc.), P-FOXO1(T24)/FOXO3A(T32) (1:1,000, Cell Signaling Technology, Inc.), P-mTOR (1:1,000, Cell Signaling Technology, Inc.), mTOR (1:1,000, Cell Signaling Technology, Inc.), P-P70S6K (1:1,000, Cell Signaling Technology, Inc.), P70S6K (1:1,000, Cell Signaling Technology, Inc.), P-4EBP1 (1:1,000, Abclonal Technology, Inc.), 4EBP1 (1:1,000, Abclonal Technology, Inc.), p62 (1:1,000, Proteintech, Inc.), LC3 (1:1,000, Sigma, Inc.), MHC (1:1,000, Developmental Studies Hybridoma Bank (DSHB)), UBC(Ubiquitin) (1:1,000, Abclonal Technology, Inc.) and GAPDH (1:10,000, Bioworld Technology, Inc.). All proteins were visualized by ECL Chemiluminescent Kit (Thermo Fisher) and chemical luminescence of membranes was detected by BioRad luminescent imaging system. Uncropped images of western blots are available in Supplementary Fig. 10.

**Stainings.** Gastrocnemius muscle samples were freshly isolated and mounted in 4% paraformaldehyde (PFA). Serial transverse sections of muscle tissues with 10 μm thickness were subjected to haematoxylin–eosin (HE) staining using commercial kit (KeyGEN, China). For PAS staining, the sections were incubated in Periodate solution, and then stained with Schiff reagent using a PAS Staining Kit following a protocol suggested by the manufacturer (Rongbio, China). For SDH staining, muscle samples were obtained, flash frozen in O.C.T Compound (optimal cutting temperature compound, Sakura) and cold isopentane, and cut at 10 μm per section. The sections were dried at room temperature for 30 min, incubated in 0.05% nitroblue tetrazolium and 0.05 M sodium succinate in 0.05 M phosphate buffer (pH 7.5) for 45 min at 37 °C, according to the instruction of SDH Staining Kit (Rongbio, China).

To determine the diameter of myotubes in vitro, C2C12 myotubes were fixed by 4% PFA for 30 min at room temperature, permeabilized with 0.5% Triton X-100 in PBS for 15 min, and then blocked with 5% BSA in PBST for 1 h at room temperature. Myotubes were incubated with anti-MHC (MF-20, 1:100, DSHB) diluted in 5% BSA overnight at 4 °C. After, myotubes were incubated with secondary antibody Cy3-AffiniPure Rabbit Anti-Mouse IgG (H + L) (1:500, Jackson) for 1 h at room temperature. Nuclear staining was performed with DAPI. Images were captured by fluorescence microscope (Leica) and the diameter of myotubes was measured by Image J.

For muscle fibre-type determination, gastrocnemius muscle samples were obtained, flash frozen in OCT and cold isopentane, and cut at 10 μm per section. The primary antibodies against MHCI (1:50, BA-F8), MHCIIa (1:50, SC-71) and MHCIIb (1:50, BF-F3) were obtained from DSHB. The corresponding secondary antibodies were obtained from Molecular Probes Thermo Fisher and listed as follows: Alexa Fluor 350 anti-mouse IgG2b (A-21140), Alexa Fluor 488 anti-mouse IgG1 (A-21121) and Alexa Fluor 555 anti-mouse IgM (A-21426). The protocol of immunofluorescent stainings for MHCI (BA-F8), MHCIIa (SC-71) and MHCIIb were the same as that for MHC. Images were captured by confocal microscope

(Zeiss) and the percentage of MHCI-, MHCIIa- and MHCIIb-positive muscle fibres to total muscle fibres was each calculated to determine muscle fibre type.

**Creatine kinase activity assay.** The creatine kinase activity in cultured medium of C2C12 myotubes was measured by ELISA (Mouse Creatine Kinase, CK ELISA Kit; Xinqidi, Wuhan, China). The culture medium of C2C12 myotubes was harvested, vortexed and centrifuged at $10,000g$ for 10 min. The assay was carried out in 96-well plates on 50 μl of 1:50 diluted samples and then the Kit was used following a standard procedure. Optical density was read at 450 nm by Microplate reader (Bio-Rad) and a standard curve was obtained with standard sample.

**miR-29b agomir injections in mice.** miR-29b agomir ($2'$OME + $5'$chol modified) and negative control agomir ($2'$OME + $5'$chol modified) (RiboBio) were used. Intramuscular injection (25 nmol per mice) was performed once a day for 3 days. After 4 days, all mice were killed and gastrocnemius muscles were removed, weighed, frozen and stored in 4% PFA. The experiments were blindly performed by the investigator who did not know the group allocation.

**Grip-strength test.** A digital grip-strength meter (YLS-13A, Yiyan Technology Co. Ltd, China) was used to measure the grip strength of mice by following a known protocol[49]. Mice were acclimatized for 10 min before the grip-strength test began. Mice were allowed to grab the metal pull bar. The force at the time of release was recorded as the peak tension. Each mouse was tested five times with a 30 s break between tests. The experiments were blindly performed by the investigator who did not know the group allocation.

**mtDNA copy number measurement.** The ratio of mtDNA to genomic DNA was calculated by dividing copies of $Co1$ with copies of $GAPDH$ in each experiment[50]. Each 10 μl reaction contained 0.5–2.0 ng of DNA extract, $1 \times$ SYBR green mix and 300 nM of each primer. Reactions were performed using a real-time PCR system: 95 °C for 10 min, followed by 50 cycles at 95 °C for 10 s, 55 °C for 15 s and 72 °C for 28 s. Fluorescence was measured during the last step of each cycle using the FAM/SYBR channel. The primers used are as followed: $Mt$-$Co1$ forward primer: $5'$-CAGTCTAATGCTTACTCAGC-$3'$, reverse primer: $5'$-GGGCAGTTA CGATAACATTG-$3'$; $GAPDH$ forward primer: $5'$-GGG AAGCCCATCAC CATCTTC-$3'$, reverse primer: $5'$-AGAGGGGCCATCCACAGT CT-$3'$.

**miR-29b sponge injections in mice.** The corresponding base pairs for miR-29b sponge regions (forward: $5'$-GATCCAACATGATTTTTTATGGTGCTACCGA ACATGATTTTTTATGGTGCTAGCGAACATGATTTTTTATGGTGCTAC-$3'$; reverse: $5'$-TCGAGTAGCACCATAAAA AATCATGTTCGCTAGCACCATAA AAAATCATGTTCGGTAGCACCATAAAAA ATCATGTTG-$3'$) for miRNA interference were designed and cloned into the FUGW cloning vector. Lentiviral particles were generated and packaged using psPAX2 and PMD2.G. A single intramuscular injection of lentiviral particles was performed at the dose of $10^8$ TU per mice. Three days after injection, denervation procedure was performed. Finally, gastrocnemius muscles were removed, weighed, frozen and stored in 4% PFA after another 5 days. The experiments were blindly performed by the investigator who did not know the group allocation.

**Statistical analysis.** Results were presented as mean ± s.e.m. An unpaired, two-tailed Student's $t$-test was used for comparisons between two groups. One-way ANOVA test was performed to compare multiple groups followed by Bonferroni's *post hoc* test. All analyses were performed using GraphPad Prism 6.0. Differences were considered significant with $P < 0.05$.

**Data availability.** Data that support the findings of this study have been deposited in Gene Expression Omnibus with the accession code GSE81914. All other relevant data are available within the article and its Supplementary Information files and from the corresponding authors on reasonable request.

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

## Acknowledgements

This work was supported by the grants from National Natural Science Foundation of China (81570362, 91639101 and 81200169 to J.X. and 81400647 to Y.B.), the development fund for Shanghai talents (to J.X.), and the National Institutes of Health (NCATS grant UH3 TR000901 to S.D.).

## Author contributions

J.X. designed the study, instructed all experiments and drafted the manuscript. J.L., M.C.C., Y.Y., Y.B., P.C., Q.Z., L.C. and O.Z. performed the experiments and analysed the data. L.C., G.C.R. and S.D. helped perform the experiments, provided technical assistance and revised the manuscript.
