## [Peer review file · Nature Communications]

Reviewers' comments:

Reviewer #1 (Remarks to the Author):

Manuscript by Li et al. demonstrated the role of miR-29 in muscle atrophy using in vitro and in vivo models. The manuscript is generally well written and presents data showing the role of miR-29 in muscle wasting. As little is still understood about the mechanisms of muscle wasting associated with a variety of factors, such as aging, diabetes or cancer, this manuscript provides interesting information for the field.

Specific comments for the authors:

- In the introduction the authors say that exercise prevents muscle atrophy - this is not necessarily correct as for example master athletes undergo muscle wasting as well despite exercising during sometimes the majority of their lives;
- The in vitro models of muscle atrophy used are not necessarily equivalent to the in vivo models therefore the manuscript seems like a collection of data rather than a "story"; perhaps re-organising some data and moving parts of it to the supplementary information may help with the "flow" of the manuscript;
- The authors used the elevated expression of Atrogin-1 and MuRF-1 as a marker of muscle atrophy - can the authors related to previous data showing downregulation of Atrogin-1 and MuRf-1 expression in muscle of older organisms (for example: Edstrom et al., 2006);
- Did the authors investigate the expression of downstream (from IGF -1) factors: AKT, SK6, mTOR (Fig.5)?
- miR-29 has been previously shown to induce muscle senescence; can the authors discuss how miR-29 can induce senescence and atrophy (shown in this manuscript)?;
- muscle wet mass is not discussed following in vivo treatments;
- there is a lack of any functional studies following miR-29 inhibition or overexpression in vivo; this could at least be discussed;
- what was the age of the rats used?
- 21 months old mice usually do not show defined sarcopenia - can the authors justify using mice at this age?
- what were the doses of Dex etc used in vivo?
- there is no information of overexpression vectors;
- what was the mutation in the 3'UTR of miRNA target genes in the luciferase constructs mutants?
- what was the dose of agomiR and antagomiR injected in vivo? How often and for how long were the injections done?
- The authors refer to satellite cells in the text - how were these isolated? Or perhaps they were primary myoblasts?
- Figure legend for Fig. 4 claims 300uM TNF α was used, whereas materials and methods state 400uM;
- N=3 is a fairly small number for in vivo experiments, especially in the aging models where the differences between individuals are significant (to the point where some propose to use frailty index rather than age) -can the authors comments on how they calculated their n numbers in experimental analysis?
- Have denervation markers been studied following denervation and miR -29 expression manipulation in vivo?
- Fig.4D - thy Y axis is not labelled;

- Fig. 5B - relative IGF1 expression is quantified from a western blot however on the membrane, there is no signal for IGF-1 in the NC inhibitor sample; similar is shown in Supplementary Fig. 2B;
- Fig. 5A - is the luciferase activity relative to Renilla luciferase activity shown?
- Fig. 6C - the muscle sections look unusual - the gaps between the fibres are quite large.
- some references relevant to the role of microRNAs in muscle wasting during aging and denervation have been omitted.

Reviewer #2 (Remarks to the Author):

The work of Jin Li and co-workers is focused on the role of miRNAs in muscle atrophy. They used a genomic approach to identify which miRNAs are induced in adult muscles of rats after denervation. A limited number of miRNAs were found to be induced and after validation by qRT-PCR only miRNA29b was confirmed to be upregulated in several catabolic conditions such as cancer, dexamethasone and absence of nutrients. The authors used in vitro experiments to sustain the biological function of miRNA29b. They performed gain and loss of function and found that miRNA29b is sufficient and required for myotube atrophy because it controls the expression levels of IGF1 and p85a-PI3K. Finally, they moved to in vivo and confirmed that miRNA29b overexpression induces atrophy while miRNA29b inhibition partially spares muscle mass after denervation. The paper is addressing an important issue about the miRNA involvement in the atrophy program. This aspect is still poorly investigated and therefore is of interest for the scientific community that is working on protein synthesis, protein degradation, signalling in muscle growth and wasting and in general for physiologist and medical doctors. However, there are several weakness in terms of signalling downstream miRNA29b and its physiological relevance. The authors should consider the following points.

Point1. Authors have shown that miR29b is induced at day 5 after denervation. Since authors expressed miR29b for several days in vivo it would be important to know the time course of miR29b expression in denervated muscles. Is it induced only at 5 days and then suppressed or is it maintained at higher levels for all the time? Authors must check miR29b expression at longer time points such as at 8 and 14 days after denervation. Authors have monitored miR29b expression in Gastrocnemius muscles. Is miR 29b upregulation specific of this muscle or is it a more generalised process? Authors must check miR29b expression in other muscles such as Soleus, EDL and Tibialis Anterior.

Point 2. It is not clear whether authors used scramble oligos as control of the miR-29b overexpression experiments shown in Fig 3a. Also authors should specify at what time of differentiation myotubes were transfected and for how long myotubes were kept in culture after transfection. How much was the increase of miR29b after the transfection? Finally authors should show that miR-29b oligos are specific and do affect other targets. Authors can use luciferase constructs containing either miR-29b or other miR target sequences to

prove the specificity.

Point3. The authors have shown that atrogin1 and MuRF1 are induced by miR29b overexpression and have found that IGF1 and PI3K are controlled by miR29b. However, neither IGF1 or PI3K directly controls atrogin1/MURF1 expression that instead is regulated by FoxO transcription factors. Authors must show/check that AKT/FoxO axis is affected by miR-29b overexpression or inhibition (Fig 5 and 6). This is particularly important for the in vivo experiments shown in Fig 6E and in Supplementary Fig 2B.

Moreover, authors must better dissect how myotubes and myofibers undergo atrophy. Is the protein ubiquitination increased by miR29b expression? Are other atrophy-related genes induced? Are other ubiquitin ligases upregulated? Is autophagy affected? The same it applies for in vivo experiments (Figure 6).

Point4. Figure 4. The authors have shown that inhibition of miR29b prevents myotube atrophy in vitro in different catabolic conditions. What happens to myotubes in basal conditions when miR29b is blocked? The expectation is that inhibition of miR29b should promote myotube growth. Moreover the authors should use the miR29b luciferase sensor to prove that miR29b was blocked by miR-29b inhibitor.

Point5. Figure 6. The in vivo experiments need more careful analyses. For instance gastrocnemius muscles shown in panel 6B looks paler than control one. Why? Is fiber type affected by miR29b expression? Please monitor Myosin expression by immunohistochemistry or Western Blots as well as mitochondria and glycogen content (SDH and PAS staining). Authors must better characterise muscle atrophy by checking protein ubiquitination, autophagy and atrogenes expression (see point3). Fibers are usually densely packed and no space is present between neighbour fibers. Panels 6C shows fibers that are abnormally separated one from the other. Are inflammation or degeneration/regeneration events occurring when miR-29b is expressed? Finally, muscle force is the best readout to prove that miR-29b is triggering a wasting condition. Authors should monitor muscle strength.

Point6. Figure 7. It is unclear why authors performed multiple injection of lentivirus vectors. Muscle is a postmitotic tissue that do not divide and lentivirus infects myofiber leading to integration of viral DNA into the genome of myonuclei. Therefore, one injection should be sufficient to transfect adult muscles. Authors must show a better image of muscle morphology to show the absence of damage. Also in this condition, fiber type should be monitored (see above point5). Inhibition of atrogin1 is detrimental in denervated muscles and causes a severe myopathy (Gomes AV et al. 2012 FASEB J; Sartori et al 2013 Nat Genet). Therefore, miR29b beneficial effects can not be explained by atrogin1/MURF1 inhibition unless other atrogenes are blocked because AKT -FoxO axis is inhibited. Authors must monitor the IGF1/AKT/FoxO axis and confirm that miR29b inhibition prevents IGF1 -Akt downregulation.

Point7. The authors have found that miR29b is induced in several catabolic conditions. It would be important to know what is triggering the upregulation of this microRNA. Is it a synergistic pathway that is controlled by the same IGF1 -Akt signalling in a feed-forward loop to enhance protein degradation? This hypothesis should be tested.

Point8. The authors did not mention and discuss a previous study that described the pattern of miRNA expression in several catabolic conditions including denervation, cancer cachexia, fasting and diabetes (Soares et al. 2014 JBC). In this study the authors identified a peculiar pattern of miRNA expression for each catabolic condition. It would be important to compare the data obtained by these two studies even if the platforms used for the microarrays experiments are different.

Reviewer #3 (Remarks to the Author):

- A. Key Results claim that miR-29b contributes to multiple types of skeletal muscle atrophy
- B. The molecule miR-29b being tested for multiple types, not a single type, of skeletal muscle atrophy is novel. However, other molecules (such as clenbuterol) have been found to retard skeletal muscle atrophy similar to miR-29b.
- C. Date & methodology are valid, approach is valid, and presentation is OK
- D. Statistics are OK
- E. Conclusions are overstated (see positives and negatives below)
- F. Need to add limb immobilization-induced skeletal muscle atrophy being tested for prevention of skeletal muscle trophy by miR-29b
- G. Literature showing incomplete prevention of skeletal muscle atrophy could be added.
- H. Manuscript is clear

For publication:

To be published in Nature Communications a paper should meet several general criteria:

1. YES, the data is technically sound
2. YES, the results are novel (we do not consider abstracts and internet preprints to compromise novelty)
3. YES, the manuscript is important to scientists in the specific field

Against publication:

To be published in Nature Communications a paper should meet several general criteria:

1. No, the paper DOES NOT provide strong evidence for its conclusions
 - a. Twice the manuscript states that exercise prevents atrophy, as follows: "Exercise is the only accepted way to prevent or attenuate muscle atrophy" - Introduction. "With the exception of exercise. There are no effective strategies to treat muscle atrophy" - Discussion. Why then was not hindlimb immobilization immobilisation used for skeletal muscles fixed by the immobilization at less than a resting length? Does miR29b prevent 100% of muscle atrophy in those muscles fixed at less than resting length during hindlimb immobilisation? Immobilization of limbs is a common clinical procedure for orthopaedic medicine.
 - b. Others have produced partial reduction of muscle atrophy with denervation before. For example, "The reduction in atrophy caused by clenbuterol treatment was particularly apparent in the larger anterior tibialis and gastrocnemius muscles (Table 1). In these muscles, denervation reduced the wet weight and protein content 61 -67% in the control rats but only H-35% in clenbuterol-treated rats. The denervated extensor digitorum longus was exceptional" in a manuscript titled; "Zeman RJ, Ludemann R, Etlinger JD. Clenbuterol, a

beta 2-agonist, retards atrophy in denervated muscles. Am J Physiol. 1987 Jan;252(1 Pt 1):E152-5. PubMed PMID: 3812670. So partial reduction of atrophy in the current reviewed manuscript is not so novel. It is only novel for the miR-29b.

Other

2. Figure 7A adjusts miR-29b with Lenti-29b sponge during denervation from its 3-fold increase in denervation fully downward to control Lenti-fugw + sham values, BUT only rescues about 30% of the skeletal muscle atrophy. Thus, the title of the manuscript ("miR-29b contributes to multiple types of atrophy") more closely approaches the truth than the heading "miR-29b controls muscle atrophy in vivo". miR-29b only PARTIALLY controls muscle atrophy in vivo., as indicated in the first sentence of this section #2.

3. Only one muscle fiber type (Type II) is tested. Susceptibility of skeletal muscle atrophy is fiber-type dependent. For example, type I muscle will atrophy 4 times more than type II muscle when limbs are immobilized so that type I skeletal muscle is fixed in a shortened position.

Reviewers' comments:

Reviewer 1:

Question 1.

In the introduction the authors say that exercise prevents muscle atrophy - this is not necessarily correct as for example master athletes undergo muscle wasting as well despite exercising during sometimes the majority of their lives.

Reply: Thank you very much for your comments. We have deleted it as suggested. Thank you very much again.

Question 2.

The in vitro models of muscle atrophy used are not necessarily equivalent to the in vivo models therefore the manuscript seems like a collection of data rather than a "story"; perhaps re-organising some data and moving parts of it to the supplementary information may help with the "flow" of the manuscript.

Reply: Thank you very much for your insightful suggestions. We have re-organized the data as suggested. Thank you very much again.

Question 3.

The authors used the elevated expression of Atrogin-1 and MuRF-1 as a marker of muscle atrophy - can the authors related to previous data showing downregulation of Atrogin-1 and MuRf-1 expression in muscle of older organisms (for example: Edstrom et al., 2006).

Reply: We have added the discussion as suggested (Page 11, Lines 15-19). Thank you very much.

Question 4.

Did the authors investigate the expression of downstream (from IGF-1) factors: AKT, SK6, mTOR (Fig.5)?

Reply: We appreciate your constructive and insightful comments. We have added the experiments as suggested. In brief, the downstream effectors (from IGF-1) were determined and we found that the phosphorylations of AKT (Ser-473), FOXO3A (Ser-253), mTOR, and P70S6K were decreased by miR-29b mimic, while all these phosphorylations were increased by miR-29b inhibitor, though the phosphorylation level of AKT (Thr-308), FOXO3A (Thr-32) and 4EBP1 was not modulated (Figure 5c-d). In addition, we found that the downstream effectors (from IGF-1) were also decreased in the atrophic gastrocnemius muscles from Den-, Dex-, and fasting-induced atrophy models (Supplemental Figure 3b and Supplemental Figure 4). Moreover, in mice injected with miR-29b agomir, expression of the downstream (from IGF-1) effectors in gastrocnemius muscles was also decreased compared to control (Supplemental Figure 5h-i). Besides that, in mice injected with miR-29b sponge in denervation, expression levels of the downstream effectors (from IGF-1) in gastrocnemius muscles were increased compared to control (Supplemental Figure 6). Thank you very much.

Question 5.

miR-29 has been previously shown to induce muscle senescence; can the authors discuss how miR-29 can induce senescence and atrophy (shown in this manuscript)?

Reply: We appreciate your constructive and insightful comments. We have already added the discussion as suggested (Page 13, Lines 17-20). Thank you very much.

Question 6.

Muscle wet mass is not discussed following in vivo treatments.

Reply: Thank you very much for your comments. We have added it. Thank you very much again.

Question 7.

There is a lack of any functional studies following miR-29 inhibition or overexpression in vivo; this could at least be discussed.

Reply: Thank you very much for your comments. We have added it as suggested. We found that miR-29b agomir led to muscle atrophy and decreased grip strength. Thank you very much again.

Question 8.

What was the age of the rats used?

Reply: The age of the rats used in this study is 7-8 weeks. We have added it as suggested. Thank you very much.

Question 9.

21 months old mice usually do not show defined sarcopenia - can the authors justify using mice at this age?

Reply: Thank you very much for pointing it out. The mice we used were 23 months old. We have corrected it. Thank you very much.

Question 10.

What were the doses of Dex etc used in vivo?

Reply: The dose of Dex used is 25mg/kg/day. We have added it in the manuscript. Thank you very much.

Question 11.

There is no information of overexpression vectors;

Reply: Thank you for your comments. We have added it as suggested. Thank you very much again.

Question 12.

What was the mutation in the 3'UTR of miRNA target genes in the luciferase constructs mutants?

Reply: We have added it as suggested. Thank you very much.

Question 13.

What was the dose of agomiR and antagomiR injected in vivo? How often and for how long were the injections done?

Reply: We have already added these information in the Methods. Thank you very much.

Question 14.

The authors refer to satellite cells in the text - how were these isolated? Or perhaps they were primary myoblasts?

Reply: They were primary myoblasts. We have corrected them. Thank you very much.

Question 15.

Figure legend for Fig. 4 claims 300uM TNFa was used, whereas materials and methods state 400uM.

Reply: Thank you very much for pointing it out. We have corrected it. Thank you very much again.

Question 16.

N=3 is a fairly small number for in vivo experiments, especially in the aging models where the differences between individuals are significant (to the point where some propose to use frailty index rather than age) -can the authors comments on how they calculated their n numbers in experimental analysis?

Reply: Thank you very much for pointing it out. For most *in vivo* experiments, we used n=4-5, and n means animal numbers from at least three independent experiments. For the aging models, n=4. For the other models, n=5. Thank you very much.

Question 17.

Have denervation markers been studied following denervation and miR-29 expression manipulation in vivo?

Reply: Thank you very much for this insightful suggestion. We have added the data as suggested. In brief, we found that although denervation significantly increased denervation markers including MuSK, AchRa, AchRe, AchRg, cPLA2, NCAM and RUNX1, miR-29b agomir only slightly elevated AchRe and cPLA2 while other denervation markers were at largely unaffected (Supplemental Figure 5f-g). Thank you very much again.

Question 18.

Fig.4D - the Y axis is not labelled.

Reply: Thank you for your comments. We have added it. Thank you very much again.

Question 19.

Fig. 5B - relative IGF1 expression is quantified from a western blot however on the membrane, there is no signal for IGF-1 in the NC inhibitor sample; similar is shown in Supplementary Fig. 2B.

Reply: We have re-performed the experiments and replaced it. Thank you very much.

Question 20.

Fig. 5A - is the luciferase activity relative to Renilla luciferase activity shown?

Reply: The luciferase activity is shown relative to Renilla luciferase activity. Thank you very much.

Question 21.

Fig. 6C - the muscle sections look unusual - the gaps between the fibers are quite large.

Reply: We have re-performed the experiments and replaced it. Thank you very much.

Question 22.

Some references relevant to the role of microRNAs in muscle wasting during aging and denervation have been omitted.

Reply: We have updated the references as suggested. Thank you very much.

Reviewer 2:

Question 1.

Authors have shown that miR29b is induced at day 5 after denervation. Since authors expressed miR29b for several days in vivo it would be important to know the time course of miR29b expression in denervated muscles. Is it induced only at 5 days and then suppressed or is it maintained at higher levels for all the time? Authors must check miR29b expression at longer time points such as at 8 and 14 days after denervation. Authors have monitored miR29b expression in Gastrocnemius muscles. Is miR 29b upregulation specific of this muscle or is it a more generalised process? Authors must check miR29b expression in other muscles such as Soleus, EDL and Tibialis Anterior.

Reply: We appreciate you constructive and insightful comments, which greatly improved our manuscript. We have added the experiments as suggested. To investigate the time course of miR-29b expression in denervated muscles, we checked its expression level at 3, 5, 7, and 14 days after denervation and found that miR-29b was induced at day 5 and maintained at higher levels after (Supplemental Figure 1f). Besides that, to explore if the up-regulation of miR-29b in denervation was specific in gastrocnemius muscles or it was a more generalized process, we checked miR-29b expression level in other muscles after denervation including tibialis anterior (TA), soleus, and extensor digitorum longus (EDL), and found that it was consistently

elevated in all these denervated muscles (Supplemental Figure 1g). Thank you very much.

Question 2.

It is not clear whether authors used scramble oligos as control of the miR-29b overexpression experiments shown in Fig 3a. Also authors should specify at what time of differentiation myotubes were transfected and for how long myotubes were kept in culture after transfection. How much was the increase of miR29b after the transfection? Finally authors should show that miR-29b oligos are specific and do affect other targets. Authors can use luciferase constructs containing either miR-29b or other miR target sequences to prove the specificity.

Reply: Thank you very much for these insightful suggestions. We used scramble oligos as control of the miR-29b overexpression experiments and we have added all other information as required. The transfection was performed after myotubes formed and 24 h later, Dex (or TNF- α or H₂O₂) was added and cells were further cultured for 24 h. miR-29b mimic increased miR-29b expression level by 157-fold, without affecting miR-29a and miR-29c (Figure 3a), which confirms that the miR-29b mimic used in this study is specific to miR-29b. Thank you very much.

Question 3.

The authors have shown that atrogen1 and MuRF1 are induced by miR29b overexpression and have found that IGF1 and PI3K are controlled by miR29b. However, neither IGF1 or PI3K directly controls atrogen1/MURF1 expression that instead is regulated by FoxO transcription factors. Authors must show/check that AKT/FoxO axis is affected by miR-29b overexpression or inhibition (Fig 5 and 6). This is particular important for the in vivo experiments shown in Fig 6E and in Supplementary Fig 2B. Moreover, authors must better dissect how myotubes and

myofibers undergo atrophy. Is the protein ubiquitination increased by miR29b expression? Are other atrophy-related genes induced? Are other ubiquitin ligases upregulated? Is autophagy affected? The same it applies for in vivo experiments (Figure 6).

Reply: Thank you very much for your comments. We have added all experiments as suggested. In brief, the downstream effectors (from IGF-1) were determined and we found that the phosphorylations of AKT (Ser-473), FOXO3A (Ser-253), mTOR, and P70S6K were decreased by miR-29b mimic while all these phosphorylations were increased by miR-29b inhibitor, though the phosphorylation level of AKT (Thr-308), FOXO3A (Thr-32) and 4EBP1 was not modulated (Figure 5c-d). In addition, we found that the downstream effectors (from IGF-1) were also decreased in the atrophying gastrocnemius muscles from Den-, Dex-, and fasting-induced atrophy models (Supplemental Figure 3b and Supplemental Figure 4). Moreover, in mice injected with miR-29b agomir, expression of the downstream effectors (from IGF-1) in gastrocnemius muscles was decreased compared to control (Supplemental Figure 5h-i). Besides that, in mice injected with miR-29b sponge in denervation, expression levels of the downstream (from IGF-1) effectors in gastrocnemius muscles were increased compared to control (Supplemental Figure 6). In fully differentiated C2C12 myotubes, we found that miR-29b overexpression decreased Foxo3 and MHC, and induced expressions of some autophagy-related genes (MAP1-LC3b, Atg7, Atg12, Bnip3, Gabarapl1, CathepsinL, Bnip3l, and Vps34) and other ubiquitin ligases-related genes (Mull1, TRAF6, ZNF216, Cblb, and Nedd4) (Figure 3c-e). *In vivo*, miR-29b agomir induced muscle atrophy, which was confirmed as evidenced by a decrease in gastrocnemius weight, gastrocnemius weight/body weight ratio, grip strength, myotube diameter, mitochondria and glycogen content (SDH and PAS staining),

MHC levels, and mtDNA copy numbers (Figure 8b-e, j, k and m); and an increase in some atrogenes including Atrogin-1, MuRF-1, MuRF2, Fbxo40, TRAF6, Cblb and Nedd4 expression, protein ubiquitination, and autophagy (Figure 8h-i, l and Supplemental Figure 5b-c). Thank you very much again.

Question 4.

Figure 4. The authors have shown that inhibition of miR29b prevents myotube atrophy in vitro in different catabolic conditions. What happens to myotubes in basal conditions when miR29b is blocked? The expectation is that inhibition of miR29b should promote myotube growth. Moreover the authors should use the miR29b luciferase sensor to prove that miR29b was blocked by miR-29b inhibitor.

Reply: Thank you very much for these suggestions. We have added the experiments as suggested. In fully differentiated C2C12 myotubes, miR-29b inhibitor was used to determine its role in regulating muscle size. miR-29b inhibitor decreased miR-29b expression level without affecting miR-29a and miR-29c (Figure 4a), which suggests that the miR-29b inhibitor used in this study is specific to miR-29b. Inhibition of miR-29b was not able to promote muscle hypertrophy in basal conditions, while it could abrogate the pro-atrophy effect of Dex stimulation (Figure 4b-c). Similarly, miR-29b inhibitor also attenuated Dex-induced atrophy in myotubes differentiated from primary myoblasts (Figure 4d). In addition, in myotubes differentiated from C2C12, treatment with TNF- α or H₂O₂ did not decrease myotube diameter and creatine kinase activity, or induce the expression of Atrogin-1 and MuRF1, when miR-29b expression was inhibited (Supplemental Figure 2c-d). Thank you very much again.

Question 5.

Figure 6. The in vivo experiments need more careful analyses. For instance

gastrocnemius muscles shown in panel 6B looks paler than control one. Why? Is fiber type affected by miR29b expression? Please monitor Myosin expression by immunohistochemistry or Western Blots as well as mitochondria and glycogen content (SDH and PAS staining). Authors must better characterise muscle atrophy by checking protein ubiquitination, autophagy and atrogenes expression (see point3). Fibers are usually densely packed and no space is present between neighbour fibers. Panels 6C shows fibers that are abnormally separated one from the other. Are inflammation or degeneration/regeneration events occurring when miR-29b is expressed? Finally, muscle force is the best readout to prove that miR-29b is triggering a wasting condition. Authors should monitor muscle strength.

Reply: Thank you very much for these suggestions. We have added the experiments as suggested. To characterize the *in vivo* relevance of overexpressing miR-29b, we used miR-29b agomir to increase the expression level of miR-29b in mouse gastrocnemius muscles. Using this approach, we could increase miR-29b level by 2.5-fold without affecting miR-29a and miR-29c (Figure 8a), with corresponding decrease in the targets as noted above (Supplemental Figure 5h-i). Muscle atrophy was confirmed as evidenced by a decrease in gastrocnemius weight, gastrocnemius weight/body weight ratio, grip strength, myotube diameter, mitochondria and glycogen content (SDH and PAS staining), MHC levels, and mtDNA copy numbers (Figure 8b-e, j, k and m); and an increase in some atrogenes including Atrogin-1, MuRF-1, MuRF2, Fbxo40, TRAF6, Cblb and Nedd4 expression, protein ubiquitination, and autophagy (Figure 8h-i, l and Supplemental Figure 5b-c). We further explored the atrophic fiber type induced by miR-29b agomir and consistently found that all types of fibers underwent atrophy as determined by SDH staining and qRT-PCR analysis of Myh7, Myh2, Myh4, Myh1 encoding myosin isoforms MyHC-I,

MyHC-IIa, MyHC-IIb, and MyHC-IIx/d (Figure 8f-g). In addition, a switch of type I fiber to type IIB fiber was observed by immunofluorescence staining with MHC antibodies (Supplemental Figure 5a). However, no obvious inflammation or degeneration/regeneration events occurred as determined by qRT-PCRs for MyoD, MyoG, Myf5, IL-1a, IL-1 β , and IL-6 (Supplemental Figure 5d-e). Besides that, although denervation significantly increased denervation markers including MuSK, AchRa, AchRe, AchRg, cPLA2, NCAM and RUNX1, miR-29b agomir only slightly elevated AchRe and cPLA2 while other denervation markers were at largely unaffected (Supplemental Figure 5f-g). Interestingly, in mice injected with miR-29b agomir, expression of IGF-1, PI3K(p85 α) and the downstream (from IGF-1) effectors in gastrocnemius muscles were decreased compared to control (Supplemental Figure 5h-i). Thus, these data indicate that the increase of miR-29 is able to induce muscle atrophy in vivo. Thank you very much.

Question 6.

Figure 7. It is unclear why authors performed multiple injection of lentivirus vectors. Muscle is a postmitotic tissue that do not divide and lentivirus infects myofiber leading to integration of viral DNA into the genome of myonuclei. Therefore, one injection should be sufficient to transfect adult muscles. Authors must show a better image of muscle morphology to show the absence of damage. Also in this condition, fiber type should be monitored (see above point5). Inhibition of atrogen1 is detrimental in denervated muscles and causes a severe myopathy (Gomes AV et al. 2012 FASEB J; Sartori et al 2013 Nat Genet). Therefore, miR29b beneficial effects can not be explained by atrogen1/MURF1 inhibition unless other atrogenes are blocked because AKT-FoxO axis is inhibited. Authors must monitor the IGF1/AKT/FoxO axis and confirm that miR29b inhibition prevents IGF1-Akt

downregulation.

Reply: Thank you for your suggestions. We re-performed the experiments using one single injection and also added other experiments as suggested. We explored the atrophied fiber type induced by miR-29b agomir and consistently found that all types of fibers underwent atrophy as determined by SDH staining and qRT-PCR analysis of Myh7, Myh2, Myh4, Myh1 encoding myosin isoforms MyHC-I, MyHC-IIa, MyHC-IIb, and MyHC-IIx/d (Figure 8f-g). In addition, a switch of type I fiber to type IIB fiber was observed by immunofluorescence staining with MHC antibodies (Supplemental Figure 5a). We provided data showing that miR29b inhibition prevents IGF1-Akt down-regulation. Thank you very much.

Question 7.

The authors have found that miR29b is induced in several catabolic conditions. It would be important to know what is triggering the upregulation of this microRNA. Is it a synergistic pathway that is controlled by the same IGF1-Akt signalling in a feed-forward loop to enhance protein degradation? This hypothesis should be tested.

Reply: We totally agree that it is important to explore the factors that triggered the up-regulation of miR-29b. To explore what triggers the up-regulation of miR-29b, we firstly investigated whether a synergistic pathway that was controlled by the same IGF-1-AKT signaling was existent in a feed-forward loop to enhance protein degradation. Knockdown of IGF-1 by siRNAs did not change the expression level of miR-29b (Figure 7a), indicating that the synergistic pathway is unlikely existent. As Yin Yang 1 (YY1) has been reported to be the upstream of miR-29b in C2C12 myoblasts, we were interested in investigating if YY1 regulated miR-29b in C2C12 myotubes. We found that knockdown of YY1 by siRNAs increased miR-29b level in C2C12 myotubes (Figure 7b). In addition, the expression level of YY1 was

consistently down-regulated in muscle atrophy induced by Den, Dex, and fasting at both mRNA and protein levels (Figure 7c-d). Importantly, in fully differentiated C2C12 myotubes, YY1 siRNAs reduced myotube diameter and elevated Atrogin-1 and MuRF1 expressions (Figure 7e-f), suggesting its functional role in regulating muscle atrophy. Thus, YY1 might probably trigger the up-regulation of miR-29b and contribute to muscle atrophy. Thank you very much.

Question 8.

The authors did not mention and discuss a previous study that described the pattern of miRNA expression in several catabolic conditions including denervation, cancer cachexia, fasting and diabetes (Soares et al. 2014 JBC). In this study the authors identified a peculiar pattern of miRNA expression for each catabolic condition. It would be important to compare the data obtained by these two studies even if the platforms used for the microarrays experiments are different.

Reply: Thank you very much for pointing it out. We have added it in the Discussion as suggested (Page 12, Lines 24-25; Page 13, Lines 1-3). Thank you very much again.

Reviewer 3:

Question 1.

Twice the manuscript states that exercise prevents atrophy, as follows: "Exercise is the only accepted way to prevent or attenuate muscle atrophy" - Introduction. "With the exception of exercise. There are no effective strategies to treat muscle atrophy" - Discussion. Why then was not hindlimb immobilization immobilisation used for skeletal muscles fixed by the immobilization at less than a resting length? Does miR29b prevent 100% of muscle atrophy in those muscles fixed at less than resting length during hindlimb immobilisation? Immobilization of limbs is a common clinical

procedure for orthopaedic medicine.

Reply: Thank you very much for your comments, which greatly improved our manuscript. As immobilization of limbs is a common clinical procedure for orthopaedic medicine, we also explored the role of miR-29b inhibition in muscle atrophy induced by immobilization of limbs. Muscle atrophy was induced by immobilization of limbs as evidenced by decreased gastrocnemius weight and gastrocnemius weight/body weight ratio, and elevated Atrogin-1 and MuRF-1, accompanied with an increase of miR-29b (Supplemental Figure 7a-c). Besides that, we also confirmed that the up-regulation of miR-29b in immobilization-induced muscle atrophy was a generalized process as it was consistently increased in TA, soleus, and EDL (Supplemental Figure 7d). A single intramuscular injection of miR-29b sponge in gastrocnemius muscle significantly inhibited miR-29b expression (Supplemental Figure 7e). Muscle atrophy was largely attenuated as evidenced by increased gastrocnemius weight and gastrocnemius weight/body weight ratio, decreased Atrogin-1 and MuRF-1, and increased muscle fiber diameter, accompanied by up-regulated expressions of IGF-1 and PI3K(p85 α) (Supplemental Figure 7f-i). These data demonstrate that suppression of miR-29b has anti-atrophy effect and could at least partly attenuate muscle atrophy.

Question 2.

Others have produced partial reduction of muscle atrophy with denervation before. For example, "The reduction in atrophy caused by clenbuterol treatment was particularly apparent in the larger anterior tibialis and gastrocnemius muscles (Table 1). In these muscles, denervation reduced the wet weight and protein content 61-67% in the control rats but only H-35% in clenbuterol-treated rats. The denervated extensor digitorum longus was exceptional" in a manuscript titled; "Zeman RJ, Ludemann R,

Etlinger JD. Clenbuterol, a beta 2-agonist, retards atrophy in denervated muscles. Am J Physiol. 1987 Jan;252(1 Pt 1):E152-5. PubMed PMID: 3812670. So partial reduction of atrophy in the current reviewed manuscript is not so novel. It is only novel for the miR-29b.

Reply: We totally agree that others have produced partial reduction of muscle atrophy with denervation before. However, this study aims at proving that miR-29b is a driver for multiple types of muscle atrophy, which is supported by the fact that miR-29b is commonly up-regulated in 5 different models of skeletal muscle atrophy including muscle atrophy induced by denervation, Dex, fasting, aging, and cancer cachexia. In addition, inhibition of miR-29b attenuated denervation and immobilization induced muscle atrophy. This study provides a novel role of miR-29b in muscle atrophy. Thank you very much.

Question 3.

Figure 7A adjusts miR-29b with Lenti-29b sponge during denervation from its 3-fold increase in denervation fully downward to control Lenti-fugw + sham values, BUT only rescues about 30% of the skeletal muscle atrophy. Thus, the title of the manuscript ("miR-29b contributes to multiple types of atrophy") more closely approaches the truth than the heading "miR-29b controls muscle atrophy in vivo". miR-29b only PARTIALLY controls muscle atrophy in vivo., as indicated in the first sentence of this section #2.

Reply: We have changed it as suggested. Thank you very much.

Question 4.

Only one muscle fiber type (Type II) is tested. Susceptibility of skeletal muscle atrophy is fiber-type dependent. For example, type I muscle will atrophy 4 times more than type II muscle when limbs are immobilized so that type I skeletal muscle is fixed

in a shortened position.

Reply: Thank you very much for this great comment. We further explored the atrophic fiber type induced by miR-29b agomir and consistently found that all types of fibers underwent atrophy as determined by SDH staining and qRT-PCR analysis of Myh7, Myh2, Myh4, Myh1 encoding myosin isoforms MyHC-I, MyHC-IIa, MyHC-IIb, and MyHC-IIx/d (Figure 8f-g). Thank you very much.

Reviewers' comments:

Reviewer #1 (Remarks to the Author):

The authors have addressed the majority of the reviewers' comments and this has improved the manuscript. However, I feel there are still some issues remaining:

- the 157-fold increase in microRNA expression following the use of miRNA mimic does not seem physiologically relevant; such an overload could potentially "clog up" RISC;
- some spelling and grammar mistakes are present;
- since the authors propose the unified role of a specific microRNA in different models of muscle atrophy, they should discuss the manuscript by Soares et al. suggesting the opposite;
- the data presented suggest that miR-29 affects atrogenes, fibre type, hypertrophy and autophagy pathways - could there be other relevant targets?
- the authors show a switch to fibre type IIb from type 1 following overexpression of miR-29b; how does this correspond to fibre type 2b being "stronger" than fibre type 1 and the observed decrease in strength phenotype?
- MHC staining in Suppl. Fig. 5A is not of good enough quality and it is not clear what the staining is for.

I believe addressing these comments would improve the quality of the manuscript.

Reviewer #2 (Remarks to the Author):

The revised version of the manuscript is greatly improved. Authors have addressed all my concerns.

Reviewer #3 (Remarks to the Author):

I have no further comments to the authors

Reviewers' comments:

Reviewer 1:

Question 1.

The 157-fold increase in microRNA expression following the use of miRNA mimic does not seem physiologically relevant; such an overload could potentially "clog up" RISC.

Reply: Thank you very much for your insightful suggestions, which greatly improved our work. To exclude the possibility that the pro-atrophy effects achieved by miR-29b mimic might not be physiologically relevant, we also used miR-29b overexpression plasmid which increased miR-29b expression by 3.39-fold, and found that myotube diameter was reduced while Atrogin-1 and MuRF-1 were elevated (Supplemental Figure 3). Thank you very much again.

Question 2.

Some spelling and grammar mistakes are present.

Reply: Thank you very much for pointing it out. We have carefully checked the manuscript to avoid the spelling and grammar mistakes. Thank you very much again.

Question 3.

Since the authors propose the unified role of a specific microRNA in different models of muscle atrophy, they should discuss the manuscript by Soares et al. suggesting the opposite.

Reply: We have added the discussion as suggested (Page 13, Lines 1-9). Thank you very much.

Question 4.

The data presented suggest that miR-29 affects atrogenes, fibre type, hypertrophy and autophagy pathways - could there be other relevant targets?

Reply: We appreciate your constructive and insightful comments. We totally agree that other relevant targets of miR-29b should be identified in the future. We have acknowledged this point as a limitation (Page 15, Lines 6-8).

Question 5.

The authors show a switch to fibre type IIb from type I following overexpression of miR-29b; how does this correspond to fibre type 2b being "stronger" than fibre type I and the observed decrease in strength phenotype?

Reply: We appreciate your constructive and insightful comments. Interestingly, we showed that the proportion of type I fiber in fiber type composition was decreased while that of type IIB fiber was increased, and also observed a decrease in strength phenotype. We think this result could be explained as follows. Firstly, all types of fibers underwent atrophy. Secondly, qRT-PCR analysis for Myh7, Myh2, Myh4, Myh1 encoding myosin isoforms MyHC-I, MyHC-IIa, MyHC-IIb, and MyHC-IIx/d showed that all myosin isoforms were decreased and the decrease of Myh7 was most significant, making the relative increase of fiber type IIB. We have added this in the Discussion (Page 13, Line 25; Page 14, Lines 1-7). Thank you very much.

Question 6.

MHC staining in Suppl. Fig. 5A is not of good enough quality and it is not clear what the staining is for.

Reply: Thank you very much for your comments. Immunofluorescent staining with MHC antibodies was used to show a switch from type I fiber to type IIb fiber. Also as suggested, we have replaced with the figure of good quality. Thank you very much.

Reviewer 2:

The revised version of the manuscript is greatly improved. Authors have addressed all

my concerns.

Reply: We appreciate your constructive and insightful comments, which greatly improved our manuscript. Thank you very much.

Reviewer 3:

I have no further comments to the authors.

Reply: Thank you very much for your comments, which greatly improved our manuscript.

REVIEWERS' COMMENTS:

Reviewer #1 (Remarks to the Author):

The revised version of the manuscript is greatly improved. Authors have addressed all my concerns.

Reviewers' comments:

Reviewer 1:

The revised version of the manuscript is greatly improved. Authors have addressed all my concerns.

Reply: Thank you very much for your insightful suggestions, which greatly improved our work.